# MUC: Machine Unlearning for Contrastive Learning with Black-box Evaluation

**Yihan Wang** *  
*University of Waterloo*  
*yihan.wang@uwaterloo.ca*

**Yiwei Lu** *  
*University of Ottawa*  
*yiwei.lu@uottawa.ca*

**Guojun Zhang** [†]  
*Alibaba*  
*guojun.zhang@uwaterloo.ca*

**Franziska Boenisch**  
*CISPA Helmholtz Center for Information Security*  
*boenisch@cispa.de*

**Adam Dziedzic**  
*CISPA Helmholtz Center for Information Security*  
*dziedzic@cispa.de*

**Yaoliang Yu**  
*University of Waterloo*  
*Vector Institute*  
*yaoliang.yu@uwaterloo.ca*

**Xiao-Shan Gao**  
*Academy of Mathematics and Systems Science, Chinese Academy of Sciences*  
*University of Chinese Academy of Sciences*  
*xgao@mmrc.iss.ac.cn*

**Reviewed on OpenReview:** *https://openreview.net/forum?id=F9pjSDvuM9*

## Abstract

Machine unlearning offers effective solutions for revoking the influence of specific training data on pre-trained model parameters. While existing approaches address unlearning for classification and generative models, they overlook an important category of machine learning models: contrastive learning (CL) methods. This paper addresses this gap by introducing the Machine Unlearning for Contrastive Learning (MUC) framework and adapting existing methods. We identify limitations in current approaches, noting that several methods perform inadequately as unlearners and that existing evaluation tools insufficiently validate unlearning effects in contrastive learning. To address these issues, we propose Alignment Calibration (AC), a novel method that explicitly considers contrastive learning properties and optimizes towards new auditing metrics for easy verification of unlearning. Through empirical comparisons with baseline methods on SimCLR, MoCo, and CLIP, we demonstrate that AC: (1) achieves state-of-the-art performance, approximating exact unlearning (retraining); (2) enables data owners to clearly visualize unlearning effects through *black-box evaluation*. The code is available at `https://github.com/EhanW/Alignment-Calibration`.

## 1 Introduction

The success of modern machine learning models largely relies on training with a large corpus of data. However, carefully annotated data are expensive and difficult to obtain, thus urging the utilization of the vast amount

---

[*]Equal contribution  
[†]Work done while at Huawei.

of unlabeled data in the wild. The recent self-supervised learning methods, especially contrastive learning methods (Chen et al., 2020; 2021; He et al., 2020), provide viable solutions to learning general representations for various downstream tasks. For example, unimodal contrastive learning models employ the InfoNCE loss to maximize the feature similarity between positive pairs (e.g., different data augmentations of the same image) while minimizing that between the negative ones (e.g., different images). This training scheme also applies to multi-modal training (e.g., CLIP (Radford et al., 2021)), and the learned encoders are widely applied in various tasks, e.g., GPT-based models (Achiam et al., 2023) and latent diffusion models (Rombach et al., 2022).

To amass large-scale datasets for training contrastive learning models, practitioners often resort to web crawling (e.g., using Common Crawl[1]). However, such data collection methods may disregard data owners' privacy concerns, potentially retrieving their data without consent. Moreover, acquired training data may include copyrighted material or even inappropriate content, such as sexual abuse (e.g., in recent reports[2] against content in LAION-5B (Schuhmann et al., 2022)). In these scenarios, data owners or authorities may rightfully request the removal of misused training data (*i.e.*, unlearning dataset) [3], necessitating adjustments to the trained model parameters. While retraining the model from scratch without the unlearning dataset is a straightforward solution, it incurs substantial computational costs for large models and datasets.

To eliminate the effect of the unlearning dataset on the model with minimum effort, machine unlearning methods (Cao & Yang, 2015; Bourtoule et al., 2021; Ginart et al., 2019; Guo et al., 2019; Neel et al., 2021; Ullah et al., 2021; Sekhari et al., 2021; Izzo et al., 2021; Chen et al., 2023; Zhang et al., 2024a; Fan et al., 2024; Shen et al., 2024) provide recipes for supervised learning methods on group removal and for generative models on sample or concept removal. However, the study of an efficient solution for contrastive learning models is under-explored[4]. In this paper, we establish the foundation of **M**achine **U**nlearning for **C**ontrastive Learning models (MUC). MUC adapts various existing methods to contrastive learning and introduces the notion of data owners who request unlearning and model owners who execute unlearning. Given candidate unlearning algorithms, the model owners first perform *white-box evaluation* to select the best method and generate an optimal unlearned model. The data owners then perform *black-box evaluation* to validate the effect of the unlearning procedure. We argue that unlearning success is achieved only if the unlearned model meets the criteria on both sides. We summarize this unlearning pipeline in Figure 1.

Unfortunately, direct adaptations of existing unlearning approaches are unsatisfactory on both considerations. Firstly, from the model owners' perspective, such algorithms are suboptimal approximations of exact unlearning (training from scratch) under different white-box evaluations and there lack of a good candidate method. Secondly, from the data owner's perspective, even given an optimal unlearned model, it is difficult to discern the unlearning effect under existing *black-box evaluation* tools, rendering it hard to determine the success of unlearning.

Motivated by the above state of affairs, we introduce a novel unlearning method called *Alignment Calibration (AC)* that is specifically tailored for contrastive learning. AC optimizes a novel loss function that involves three terms: (1) a positive alignment calibration term that removes the footprint of the unlearn set on the representation; (2) a negative alignment calibration term that leaves explicit unlearning traces for evaluation; (3) a performance preserving term that maintains uniformity.

Finally, we empirically compare baseline methods with our *Alignment Calibration* algorithms on unlearning models pre-trained on SimCLR (Chen et al., 2020) MoCo (He et al., 2020), and CLIP (Radford et al., 2021). Under various unlearning settings (e.g., the fraction of the unlearning dataset) and evaluation metrics, *AC* consistently outperforms the baseline methods, especially under unlearn evaluation, validating the benefits of our method. In summary, we make the following contributions:

---

[1]`https://commoncrawl.org/`

[2]`https://purl.stanford.edu/kh752sm9123`

[3]In accordance with policies such as the European Union's General Data Protection Regulation (GDPR), the California Consumer Privacy Act (CCPA), and Canada's proposed Consumer Privacy Protection Act (CPPA).

[4]We note that Wang et al. (2023) addresses a related problem in contrastive unlearning. However, it differs significantly from our approach in several key aspects. Notably, Wang et al. (2023) does not provide a comprehensive evaluation framework or establish strong baseline comparisons.

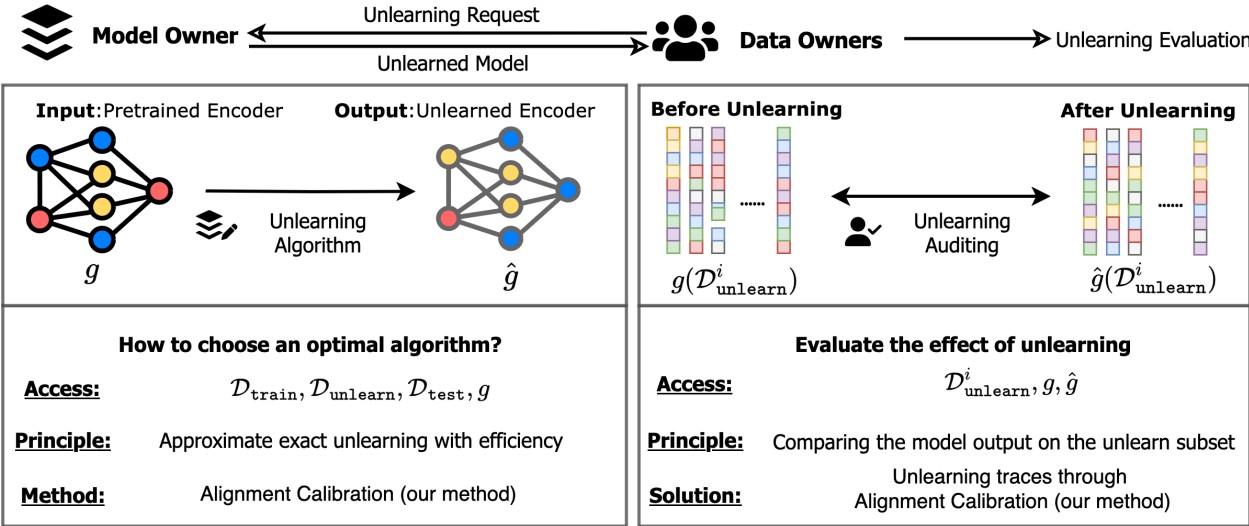

Figure 1: An overview of the unlearning pipeline for contrastive learning. Given a training set $\mathcal{D}_{\texttt{train}}$ contains an unlearn subset $\mathcal{D}_{\texttt{unlearn}}$, and a pretrained encoder $g$, model owners (Column 1) aim to find an optimal unlearning algorithm by comparing with exact unlearning (retraining with the retain dataset $\mathcal{D}_{\texttt{train}} \backslash \mathcal{D}_{\texttt{unlearn}}$) to obtain an unlearned encoder $\hat{g}$. Data owners (Column 2), who only have *black-box* access to the model outputs, then evaluate the unlearning outcome by comparing the output features before/after unlearning.

- We propose the MUC framework that considers existing methods and various evaluation tools in contrastive learning, including *white-box evaluation* and *black-box evaluation*.

- Motivated by insufficiencies of existing unlearning algorithms and evaluation tools, we propose the novel *Alignment Calibration* method that satisfies both model owners and data owners.

- Our experiments initiate the evaluation of existing machine unlearning methods for contrastive learning and confirm the superiority of our new methods.

## 2 Background and Related Work

We first provide background and related work on contrastive learning and machine unlearning.

**Contrastive Learning and Self-supervised Learning**   Contrastive learning learns general representations by contrasting sample pairs, which analytically benefits the downstream applications (Saunshi et al., 2019; Tosh et al., 2021). Popular contrastive learning methods such as Contrastive Predictive Coding (CPC) (van den Oord et al., 2018), SimCLR (Chen et al., 2020), and MoCo (He et al., 2020) employ the InfoNCE loss to enforce the contrast between positive and negative pairs. Other variants of the InfoNCE-based loss are also widely applied, e.g., $f$-MICL (Lu et al., 2023), Alignment and Uniformity (Wang & Isola, 2020), and Pearson $\chi^2$ divergence (Tsai et al., 2021). This contrastive training scheme is also applied to the context of multimodal learning, where images and texts are formed as pairs, e.g., in CLIP (Radford et al., 2021). There exist other self-supervised learning methods that also learn representations (Grill et al., 2020; Chen & He, 2021; He et al., 2022; Caron et al., 2021). In this paper, we mainly focus on developing unlearning recipes for contrastive learning methods, especially SimCLR, MoCo, and CLIP.

Specifically, contrastive learning usually applies the InfoNCE loss to learn a representation $g$. Given a probability measure $p$, we define the density of *positive pairs* sampled from $p$ as $p^+$, *i.e.*, two samples with similar feature embeddings as the joint distribution. Specifically, one minimizes the loss below as the objective:

$$\mathcal{L}_{\texttt{InfoNCE}} = - \mathbb{E}_{(x,y)\sim p^+} s(g(x), g(y)) + \mathbb{E}_{x\sim p} \log \mathbb{E}_{y\sim p} \exp\big((s(g(x), g(y))\big), \tag{1}$$

where $s$ is the cosine similarity after normalization with a temperature parameter, and $g(x), g(y)$ are the features extracted by a given encoder $g$,[5] respectively. The above contrastive learning (pre-training) scheme learns a general encoder $g$ (image and text encoders for CLIP). Such a (fixed) $g$ can be utilized with an additional linear head or shallow models for downstream tasks. In this paper, we mainly consider linear probing, where $g$ is used for the classification of the same dataset with pretraining. Notably, we consider unlearning during the pretraining phase only.

**Machine Unlearning** *For Supervised Learning*: Machine unlearning (MU) (Cao & Yang, 2015) requires an algorithm to revert to a state that specific data points are never trained on. While exact unlearning (Bourtoule et al., 2021) (e.g., retraining the model entirely on the retain dataset) provides a reliable solution, the additional computation requirement is also tremendous. In this paper, we focus on **approximate unlearning** (Ginart et al., 2019; Guo et al., 2019; Neel et al., 2021; Ullah et al., 2021; Sekhari et al., 2021; Izzo et al., 2021; Chen et al., 2023; Zhang et al., 2024a; Fan et al., 2024; Shen et al., 2024) to efficiently achieve the same goal.

*For Generative Models:* MU methods are applied to diffusion models to avoid copyright infringement and inappropriate image generation (Gandikota et al., 2023; Zhang et al., 2023b; Heng & Soh, 2023; Kumari et al., 2023). For large language models, MU is applied as a model-editing (Yao et al., 2023) tool to enable forgetting on certain training texts (Mitchell et al., 2022b;a; Jang et al., 2022; Eldan & Russinovich, 2023; Zhang et al., 2023a; Hu et al., 2024; Jia et al., 2024; Maini et al., 2024; Liu et al., 2024; Zhang et al., 2024b). In this paper, we focus on MU on self-supervised learning, specifically, contrastive learning methods, which differs from the above two cases in both unlearning settings and frameworks, which we specify in the following section.

## 3 Machine Unlearning for Contrastive Learning (MUC)

In this section, we specify the problem setting of machine unlearning for contrastive learning, introduce direct adaptations of existing methods, and propose evaluation metrics.

### 3.1 Problem Settings

**Formal Notations:** (1) We denote the pretrained encoder as $g$ and the encoder after unlearning as $\hat{g}$. Given an input sample $x$, the features extracted by the encoders are denoted as $g(x)$ and $\hat{g}(x)$ respectively; (2) We denote the original training set as $\mathcal{D}_{\texttt{train}}$ (which is used to train $g$), the unlearn set as $\mathcal{D}_{\texttt{unlearn}}$ and the retain set as $\mathcal{D}_{\texttt{retain}} = \mathcal{D}_{\texttt{train}} \backslash \mathcal{D}_{\texttt{unlearn}}$ ($\backslash$ denotes removal here); (3) We define two parties involved in unlearning: the model owner ⬙ who receives the unlearning request, and the data owner ⬤ who wishes to remove data. In practical scenarios, ⬤⬤ consist of a group of individuals $\{⬤^i\}_{i=1}^N$, who may not know the existence of each other, but participate in unlearning at the same time.

**Approximate Unlearning:** The model owner ⬙ aims at choosing an unlearning algorithm that approximates exact unlearning (training on $\mathcal{D}_{\texttt{retain}}$ from scratch) to obtain $\hat{g}$. In the meantime, this algorithm should be much more efficient than exact unlearning. Given a pool of candidate algorithms, ⬙ performs *white-box evaluations* (access to $\mathcal{D}_{\texttt{train}}, \mathcal{D}_{\texttt{unlearn}}, \mathcal{D}_{\texttt{test}}, g$ and candidates $\{\hat{g}\}$) by comparing their performances with exact unlearning according to metrics in Section 3.3.

**Unlearning evaluation:** Assuming the unlearning process is finished and the model owner has published the final unlearned model $\hat{g}$, an individual data owner $⬤^i$ wishes to examine the unlearning outcome. We coin this process as **unlearning evaluation**. Specifically, such evaluation is *black-box* as $⬤^i$ only has access to his/her own unlearning subset $\mathcal{D}_{\texttt{unlearn}}^i$, and the output of the model before $g(\mathcal{D}_{\texttt{unlearn}}^i)$ and after unlearning $\hat{g}(\mathcal{D}_{\texttt{unlearn}}^i)$. By comparing these outputs, $⬤^i$ should find explicit evidence that unlearning has indeed been performed and the output is as desired. Our paper aims to provide such evidence through the design of a novel unlearning algorithm for contrastive learning.

---

[5]Note that in practice, $g$ may consist of a general encoder and a projection head (which is typically removed for downstream tasks), as in Chen et al. (2020). For simplicity, we use $g$ to represent this entire mapping.

Note that making unlearning easy to evaluate is an important and difficult task for approximate unlearning algorithms (Thudi et al., 2022). While we provide easy-to-check unlearning traces in the later sections, such tools are specifically designed for our algorithm in contrastive learning, and may not be generalized to other unlearning scenarios.

## 3.2 Adapting existing methods to MUC

We first adapt some existing unlearning methods designed for supervised unlearning to contrastive unlearning. Due to the lack of labels in contrastive learning pre-training, some approaches cannot be directly applied. For example, random labeling (Golatkar et al., 2020; Fan et al., 2024) relies on flipping the labels of the unlearn data; boundary unlearning (Chen et al., 2023) expands or shrinks the decision boundary, which does not exist in our context. In contrast, some other unlearning methods can be tailored to contrastive learning. Specifically, we adapt the following methods:

- *Retraining*: (exact unlearning) trains on $\mathcal{D}_{\texttt{retain}}$ from scratch via minimizing Equation (1);

- *Fine-tuning* (Golatkar et al., 2020) updates the pre-trained model for several epochs on $\mathcal{D}_{\texttt{retain}}$;

- *Gradient Ascent* (Golatkar et al., 2020; Neel et al., 2021; Thudi et al., 2022) reversely maximizes Equation (1) on the $\mathcal{D}_{\texttt{unlearn}}$;

- *NegGrad* (Kurmanji et al., 2023) jointly minimizes and maximizes Equation (1) on $\mathcal{D}_{\texttt{retain}}$ and $\mathcal{D}_{\texttt{unlearn}}$ respectively;

- $\ell_1$-*Sparsity* (Jia et al., 2023) regularizes the $\ell_1$-norm of model parameters based on fine-tuning.

The above methods manifest straightforward adaptations of unlearning from supervised learning to contrastive learning by changing the cross entropy loss to the InfoNCE loss in Equation (1). In Section 5, we show that approximate unlearning methods are suboptimal approximations of exact unlearning, namely that there still exists a performance gap compared with retraining. This motivates us to design new unlearning methods specifically for contrastive learning in Section 4.

## 3.3 How to choose an unlearning algorithm

Suppose the model owner ⧉ gathers a pool of unlearning algorithms (e.g., the methods above). Next we introduce how to compare them, *i.e.*, the evaluation metrics. As contrastive learning returns a feature extractor, we can either evaluate unlearning on the representations directly or rely on downstream tasks. Specifically:

- *Representation-level metrics.* ❶ Forgetting Score: given a candidate algorithm that updates $g$ to $\hat{g}$, we propose a Forgetting Score (FS) by directly adapting the memorization score in evaluating data attribution in Wang et al. (2024). FS measures the quantity of forgetting $\mathcal{D}_{\texttt{unlearn}}$ by comparing the alignment loss through the features returned by model parameters before and after unlearning:

$$\texttt{FS} := \mathop{\mathbb{E}}_{(x,y)\sim p_{\texttt{u}}^+} s(g(x), g(y)) - \mathop{\mathbb{E}}_{(x,y)\sim p_{\texttt{u}}^+} s(\hat{g}(x), \hat{g}(y)), \tag{2}$$

where $p_{\texttt{u}}$ is the density of $\mathcal{D}_{\texttt{unlearn}}$, recall $g$ and $\hat{g}$ are models before/after unlearning.
❷ Membership Inference Attacks (MIA): MIAs are capable of indicating whether certain samples (e.g., the unlearn set) are included in the training set and are perfect for examining unlearning efficacy. EncoderMI (Liu et al., 2021) proposed an alignment-based membership inference attack for self-supervised encoders. It extracts membership information from the embedded features to distinguish whether input data is included in the encoder training set. Following the implementation of Jia et al. (2023); Fan et al. (2024), we evaluate the attack success rate (ASR) on the unlearn dataset $\mathcal{D}_{\texttt{unlearn}}$ and denote it by encoder membership inference attack (**EMIA**) efficacy. We compare EMIA on $\mathcal{D}_{\texttt{unlearn}}$ with retraining. See Appendix A.3 for details of EMIA.

- *Downstream-level metrics.* Alternatively, representations can be evaluated with downstream tasks. We perform linear probing, *i.e.*, image classification on the same (labeled) dataset for unimodal contrastive learning. Given the unlearned encoder $\hat{g}$ returned by a candidate algorithm, we train an additional linear head on $\mathcal{D}_{\texttt{retain}}$ on top of the fixed $\hat{g}$ to obtain a classifier. Next we evaluate: ❶ Accuracies: we evaluate retain accuracy (**RA**) on $\mathcal{D}_{\texttt{retain}}$, test accuracy on $\mathcal{D}_{\texttt{test}}$ (**TA**), and unlearn accuracy on $\mathcal{D}_{\texttt{unlearn}}$ (**UA**). For a good unlearning algorithm, the above three measurements should be close to those of the retrained model, with a common pattern of $\mathbf{UA} \approx \mathbf{TA} < \mathbf{RA}$; ❷ Membership Inference Attacks: Similarly to EMIA, we implement a confidence-based membership inference attack (Jia et al., 2023; Fan et al., 2024; Song & Mittal, 2021) on the entire network (encoder and linear head) and report classifier membership inference attack (**CMIA**) efficacy. We compare CMIA with retraining. See Appendix A.3 for details.

### 3.4 How to evaluate unlearning

After choosing an optimal unlearning algorithm, the model owner generates the unlearned model $\hat{g}$ as a response to the unlearning request made by data owners. However, it is impossible for the data owners to perform the same *white-box evaluations*.

**For the data owners 👥 (Unlearning evaluation) :**  Recall that an individual data owner 👤$^i$ performs *black-box evaluation* due to the limited access to input $\mathcal{D}^i_{\texttt{unlearn}}$ and the output of the encoder before/after unlearning. Specifically, 👤$^i$ cannot train shadow[6] models with $\mathcal{D}^i_{\texttt{unlearn}}$ alone to perform MIAs; and cannot obtain **TA** or **RA** to quantify performance. Additionally, there is a lack of the retrain baseline to compare with. To this end, the only evaluation tool is the forgetting score $\texttt{FS}$ on $\mathcal{D}_{\texttt{unlearn}}$, which can be calculated with Equation (2). However, we argue that this evaluation is ***neither sufficient nor reliable***, and we use a simple empirical example to validate this claim:

*Exact unlearning on MoCo* (He et al., 2020): Using a fixed random seed of 11, we pretrain a MoCo encoder $g_1$ on the entire CIFAR-10 training set. For exact unlearning, we retrain an encoder $g_2$ after removing 10% of the data, with the same random seed. To simulate a scenario where the model owner attempts to deceive the unlearning process, we assume that instead of retraining, the original encoder $g_1$ is replaced with another encoder $g_3$ that is pre-trained on the entire training data with a different random seed of 10. In this setting, the null hypothesis $H_0$ (no unlearning) corresponds to the deceptive case in which $g_1$ is replaced by $g_3$. The alternative hypothesis $H_1$ represents the exact unlearning case where $g_1$ is replaced by $g_2$. For both hypotheses, we calculate the $\texttt{FS}$ for every unlearn sample and get the mean $\mu$ and the standard deviation $\sigma$ across the 4500 unlearning images. The resulting statistics are: $H_0 : (\mu_0 = -0.0026, \sigma_0 = 0.0587)$; $H_1 : (\mu_1 = 0.0353, \sigma_1 = 0.0575)$. Assuming Gaussian distributions and a data owner possesses $N$ images, we perform a t-test to evaluate the $p$-value for distinguishing between $H_0$ and $H_1$:

Table 1: $p$-value of the t-test on distinguishing between hypothesis $H_0$ and $H_1$.

| $N$ | **5** | **10** | **15** | **20** |
|---|---|---|---|---|
| $p$-value | 0.3322 | 0.1617 | 0.0847 | 0.0459 |

In Table 1, we observe that when the data owner holds fewer than 15 images, it becomes difficult to statistically detect that the model owner has cheated the unlearning process.

In summary, under the current MUC framework, existing approximate unlearning algorithms and evaluation tools are insufficient. To address this, we design new unlearning algorithms for the model owners and advanced evaluation tools for data owners in contrastive learning in the next section, which would benefit both parties in engaging the unlearning procedure.

---

[6]Surrogate models to train classifiers for membership inference.

## 4 Alignment Calibration

### 4.1 Tailored objective for MUC

We first introduce a more effective unlearner for model owners ⬙ . Recall that ⬙ 's goal for unlearning is preserving the model utility on $\mathcal{D}_{\texttt{retain}}$ while revoking the effects of training on $\mathcal{D}_{\texttt{unlearn}}$. For the retain dataset $\mathcal{D}_{\texttt{retain}}$, we minimize the InfoNCE loss in Equation (1) to achieve reasonable downstream performance after unlearning:

$$\mathcal{L}_{\texttt{retain}} = - \mathop{\mathbb{E}}_{(x,y)\sim p_{\texttt{r}}^+} s(g(x), g(y)) + \mathop{\mathbb{E}}_{x\sim p_{\texttt{r}}} \log \mathop{\mathbb{E}}_{y\sim p_{\texttt{d}}} \exp(s(g(x), g(y)), \tag{3}$$

where $p_{\texttt{r}}$ is the density of $\mathcal{D}_{\texttt{retain}}$ and $p_{\texttt{d}}$ is the density of $\mathcal{D}_{\texttt{train}}$.

For the unlearn dataset $\mathcal{D}_{\texttt{unlearn}}$, revoking the effects of training amounts to achieving the following goals upon evaluation in Section 3.3:

- (*Encoder-level*) Enlarging forgetting on $\mathcal{D}_{\texttt{unlearn}}$ : recall that in Equation (2) the forgetting score FS is measured by the difference between feature similarity on $\mathcal{D}_{\texttt{unlearn}}$ before/after unlearning with pre-trained model $g$ and unlearned model $\hat{g}$. As the first term is fixed (as $g$ is given) during unlearning, increasing FS is equal to minimizing the second positive alignment term. For this purpose, we explicitly perform such minimization in our objective function and call it *positive alignment calibration.*

- (*Downstream-level*) $\mathbf{UA} \approx \mathbf{TA} < \mathbf{RA}$: enlarging FS alone may also hurt the overall downstream performance on the unlearned model $\hat{\mathbf{g}}$. To obtain reasonable UA and TA, we find it beneficial to maintain the term for negative pairs in contrastive learning, such that for $\mathcal{D}_{\texttt{unlearn}}$, we minimize:

$$\mathcal{L}_{\texttt{unlearn}} = \underbrace{\mathop{\mathbb{E}}_{(x,y)\sim p_{\texttt{u}}^+} s(g(x), g(y))}_{positive\ alignment\ calibration} + \underbrace{\mathop{\mathbb{E}}_{x\sim p_{\texttt{u}}} \log \mathop{\mathbb{E}}_{y\sim p_{\texttt{d}}} \exp(s(g(x), g(y)))}_{performance\ preserving}, \tag{4}$$

where $p_{\texttt{u}}$ is the density of $\mathcal{D}_{\texttt{unlearn}}$. Wang & Isola (2020) states that alignment and uniformity are crucial properties of good representations. While we calibrate alignment with the first term, our performance preserving term implicitly maintains uniformity, which we will demonstrate in Section 5.5.

### 4.2 Calibration under unlearning evaluation

**Evaluation beyond FS:** Recall that in Section 3.3, we show that the forgetting score FS is not a sufficient nor reliable evaluation for unlearning success. Here we introduce an additional evaluation tool: given $\mathcal{D}_{\texttt{unlearn}}$ and the models before unlearning $g$, data owners 👥 can easily obtain the feature vectors with two different data augmentations: $g(\mathbf{x}) = \{g(x_i)\}_{i=1}^{|\mathcal{D}_{\texttt{unlearn}}|}$ and $g(\mathbf{y}) = \{g(y_j)\}_{j=1}^{|\mathcal{D}_{\texttt{unlearn}}|}$. Then an `Alignment Matrix` : $\texttt{AM}(g(\mathbf{x}), g(\mathbf{y}))$ can be easily acquired by calculating the pairwise similarity between the two vectors. See Figure 3(a) for some visualizations of AM in the format of heatmaps. Similarly, 👥 can obtain $\texttt{AM}(\hat{g}(\mathbf{x}), \hat{g}(\mathbf{y}))$ after unlearning and an additional `Alignment Gap Matrix` : $\texttt{AGM} = \texttt{AM}(g(\mathbf{x}), g(\mathbf{y})) - \texttt{AM}(\hat{g}(\mathbf{x}), \hat{g}(\mathbf{y}))$ . The heatmaps of AM and AGM provide evaluation tools beyond FS and allow 👥 to visualize the model change through unlearning by looking at the temperature of the graphs. Notably, the elements on the diagonal of AGM also visualize sample-wise forgetting scores. We direct readers to visualizations of such graphs in Figure 2 and Figure 3 in Section 5.

**Taking evaluation into account for unlearning:** The additional evaluation tools enable the model owners to design an algorithm that allows data owners to clearly visualize the effect caused by unlearning (*i.e.*, through AM or AGM) without sacrificing the goal of unlearning.

We provide a simple solution to improve existing unlearning methods. As we have explicitly calibrated the alignment of positive pairs of $\mathcal{L}_{\texttt{unlearn}}$ in Equation (4), it suffices to adjust that of negative pairs (within $\mathcal{D}_{\texttt{unlearn}}$) to a larger value to enlarge the model differences in AM. Specifically, we update the unlearn loss in

Equation (4) with *negative alignment calibration*:

$$-\alpha \cdot \underbrace{\mathbb{E}_{(x,y)\sim p_{\mathrm{u}}^{\times}} s(g(x),g(y))}_{\textit{negative alignment calibration}} + \beta \cdot \underbrace{\mathbb{E}_{(x,y)\sim p_{\mathrm{u}}^{+}} s(g(x),g(y))}_{\textit{positive alignment calibration}} + \gamma \cdot \underbrace{\mathbb{E}_{x\sim p_{\mathrm{u}}} \log \mathbb{E}_{y\sim p_{\mathrm{d}}} \exp(s(g(x),g(y)))}_{\textit{performance preserving}},$$

where $\alpha, \beta, \gamma$ are tunable parameters to adjust the strength of each component, $p_u^{\times}$ represents negative pairs within the negative set. We write the complete objective of *Alignment Calibration (AC)*:

$$\mathcal{L}_{\mathtt{retain}} + \varepsilon \cdot \mathcal{L}_{\mathtt{unlearn}}, \tag{5}$$

where $\varepsilon = |\mathcal{D}_{\mathtt{unlearn}}|/|\mathcal{D}_{\mathtt{retain}}|$ varies by the size of the unlearn set. In the next section, we will show *AC* not only achieves state-of-the-art performance upon model owners' evaluations but can also easily pass data owners' visual evaluation on unlearning.

## 5 Experiment

Recall that we made several claims in Section 3 and Section 4: ❶ Existing methods are suboptimal unlearners under *white-box evaluations* and our *AC* algorithm approaches exact unlearning in this regard; ❷ Under MUC, *AC* introduces alignment matrices AM for visual evaluation and exhibits clear evidence for unlearning. In this section, we evaluate the baseline methods and *AC* following the above steps and present *white-box evaluation* by model owners ⧨ and *black-box evaluation* by data owners 👥 .

### 5.1 Experimental Setup

**Data and models.**    For unimodal contrastive unlearning, we perform experiments on CIFAR-10/CIFAR-100 Krizhevsky et al. (2009)/SVHN (Netzer et al., 2011) and SimCLR (Chen et al., 2020)/MoCo (He et al., 2020) algorithms with the ResNet-18 (He et al., 2016) backbone (we provide additional results on ResNet-50 in Table 12 in Appendix B). We randomly forget 10/50% training data from a pre-trained encoder. For multimodal contrastive unlearning, we evaluate CLIP (Radford et al., 2021) on an Image-Text paired dataset called MS-COCO (Lin et al., 2014), which contains ∼120K images and ∼600K captions. We perform unlearning on 10% randomly selected image-text pairs.

**White-box Evaluation:**    Following Section 3.3, we use FS and EMIA for encoder-level evaluation, and use CMIA, RA, TA, and UA for downstream-level evaluation after performing linear probing for SimCLR/MoCo experiments. For the evaluation of CLIP, we measure the image-text cosine similarity of the retain dataset and unlearn dataset due to the lack of suitable downstream tasks. Across all experiments, we compare each unlearning method with the exact unlearning (retraining) baseline and report the differences across all metrics. We also report the running time efficiency (RTE) of unlearning methods to evaluate efficiency.

**Black-box evaluation:**    Recall that due to the limited access of data owners, the above white-box evaluation can not be directly applied. Instead, we use the Alignment Matrix (AM) and Alignment Gap Matrix (AGM) introduced in Section 4 for visual evaluation on MoCo and CLIP.

**Unlearning Algorithms:**    We evaluate Retrain, Fine-Tune, Gradient Ascent, NegGrad, and $\ell_1$-Sparsity as baselines for MUC. Our *Alignment Calibration* method updates the pre-trained encoder for the same number of epochs as FineTune, NegGrad, and $\ell_1$-Sparsity. For simplicity, we set $\alpha = \gamma = 1$ if not otherwise stated and we tune $\beta$ for the best performance. Implementation details of the above methods are described in Appendix A.2.

### 5.2 Unlearning Performance under White-box Evaluation

We first provide empirical evidence for model owners to choose a suitable unlearning method with superb efficiency and effectiveness. ❶ Unimodal contrastive learning: We present our evaluation under EMIA, RA, TA, UA and CMIA in Table 2 and Table 3 for unlearning 10/50% of CIFAR-10 training set and FS (CIFAR-10,

Table 2: Unlearning performance of different methods on randomly forgetting 10% of CIFAR-10 training data under various metrics. The performance gaps between retraining and other methods are shown in the parenthesis. We report the average gap (Avg. Gap) over these 5 metrics. The results are obtained by averaging over 5 random trials.

| Methods | EMIA | RA | TA | UA | CMIA | Avg. Gap (%) ↓ | RTE (mins) ↓ |
|---|---|---|---|---|---|---|---|
| | | | MoCo | | | | |
| Retrain | 49.72 | 89.54 | 87.76 | 88.42 | 34.38 | - | 109.47 |
| Fine-Tune | 50.15 (0.43) | 88.34 (1.20) | 86.46 (1.30) | 87.59 (0.83) | 29.42 (4.96) | 1.74 | 1.42 |
| Grad. Ascent | 44.95 (4.77) | 89.92 (0.38) | 88.28 (0.52) | 89.76 (1.34) | 28.53 (5.85) | 2.67 | 0.17 |
| NegGrad | 48.43 (1.29) | 89.25 (0.29) | 87.35 (0.40) | 88.58 (0.16) | 28.89 (5.49) | 1.53 | 1.70 |
| $\ell_1$-Sparsity | 49.38 (0.34) | 88.56 (0.98) | 86.91 (0.84) | 88.12 (0.30) | 29.91 (4.47) | 1.39 | 1.43 |
| AC (Ours) | 50.28 (0.56) | 89.14 (0.40) | 87.24 (0.52) | 88.20 (0.22) | 31.50 (2.88) | **0.92** | 1.87 |
| | | | SimCLR | | | | |
| Retrain | 48.11 | 90.87 | 88.94 | 89.68 | 38.87 | - | 151.77 |
| Fine-Tune | 47.72 (0.39) | 89.38 (1.49) | 87.26 (1.68) | 88.93 (0.75) | 30.71 (8.16) | 2.49 | 1.93 |
| Grad. Ascent | 41.48 (6.63) | 91.26 (0.40) | 89.55 (0.61) | 91.11 (1.43) | 29.36 (9.50) | 3.71 | 0.19 |
| NegGrad | 49.56 (1.46) | 89.10 (1.77) | 87.23 (1.71) | 89.07 (0.61) | 29.97 (8.89) | 2.89 | 2.34 |
| $\ell_1$-Sparsity | 48.44 (0.33) | 90.59 (0.28) | 88.56 (0.38) | 90.44 (0.75) | 30.61 (8.26) | 2.00 | 1.96 |
| AC (Ours) | 48.64 (0.53) | 90.24 (0.63) | 88.06 (0.88) | 89.24 (0.44) | 33.12 (5.75) | **1.65** | 3.00 |

Table 3: Unlearning performance of various methods on randomly forgetting 50% of CIFAR-10 training data. The results are averaged over 5 random trials.

| Methods | EMIA | RA | TA | UA | CMIA | Avg. Gap (%) ↓ | RTE (mins) ↓ |
|---|---|---|---|---|---|---|---|
| | | | MoCo | | | | |
| Retrain | 55.95 | 85.98 | 83.55 | 83.98 | 46.66 | - | 66.71 |
| Fine-Tune | 49.98 (5.97) | 87.89 (1.90) | 85.66 (2.12) | 86.65 (2.67) | 32.35 (14.31) | 5.39 | 0.86 |
| Grad. Ascent | 42.90 (13.06) | 89.51 (3.53) | 87.79 (4.25) | 88.99 (5.01) | 31.85 (14.82) | 8.13 | 0.45 |
| NegGrad | 57.40 (1.45) | 83.04 (2.94) | 80.15 (3.40) | 80.59 (3.39) | 40.65 (6.02) | 3.44 | 1.66 |
| $\ell_1$-Sparsity | 52.19 (3.76) | 81.60 (4.38) | 80.19 (3.36) | 80.77 (3.20) | 38.43 (8.24) | 4.59 | 0.87 |
| AC (Ours) | 55.02 (0.93) | 86.28 (0.30) | 83.28 (0.26) | 83.72 (0.26) | 38.39 (8.27) | **2.00** | 1.84 |
| | | | SimCLR | | | | |
| Retrain | 53.37 | 87.23 | 85.16 | 85.69 | 49.30 | - | 89.74 |
| Fine-Tune | 45.90 (7.47) | 87.88 (0.65) | 85.50 (0.34) | 87.23 (1.54) | 35.44 (13.85) | 4.77 | 1.17 |
| Grad. Ascent | 42.23 (11.13) | 90.52 (3.28) | 88.61 (3.45) | 90.45 (4.77) | 33.28 (16.02) | 7.73 | 0.59 |
| NegGrad | 55.70 (2.33) | 83.98 (3.25) | 82.15 (3.01) | 83.80 (1.89) | 33.67 (15.62) | 5.22 | 2.32 |
| $\ell_1$-Sparsity | 46.51 (6.86) | 89.84 (2.60) | 87.75 (2.69) | 89.48 (3.80) | 35.38 (13.91) | 5.95 | 1.19 |
| AC (Ours) | 47.12 (6.25) | 86.11 (1.13) | 83.92 (1.24) | 85.24 (0.45) | 37.57 (11.72) | **4.16** | 3.07 |

MoCo) for 10/50% separately in Table 8 due to different scales. We also report the average gap percentage in Appendix B.3. For both SimCLR and MoCo, our proposed *Alignment Calibration (AC)* method achieves the lowest average performance gap over EMIA, RA, TA, UA, and CMIA. In terms of unlearning efficiency, our method only introduces a slight overhead. Additionally, our method achieves the lowest `FS` gap compared to retraining. In Table 4 and Table 5, we also report the results on CIFAR-100, where our methods consistently achieve the best performance. In Table 6, we report the unlearning results on the SVHN dataset. Our Alignment Calibration method achieves the least average gap to Retraining compared to the other 4 baseline methods while showing comparable efficiency in the unlearning process. ❷ Multi-modal contrastive learning: in Table 7, we again observe that our method is the best approximator of exact unlearning when evaluating the image-text cosine similarity.

## 5.3 Black-box evaluation

Motivated by the insufficiency of evaluation with the `FS` score, we propose to apply the `Alignment Matrix (AM)` and `Alignment Gap Matrix (AGM)` in Section 4. `AM` and `AGM` naturally introduce additional quantification of negative alignment. In Figure 2, we report the negative alignment value (mean and standard deviation of pairwise similarity on negative samples in `AGM`) of 4500 unlearn samples and observe that our method *AC*

Table 4: Unlearning performance of various methods on randomly forgetting 10% of CIFAR-100 training data. The results are averaged over 5 random trials.

| Methods | EMIA | RA | TA | UA | CMIA | Avg. Gap (%) ↓ | RTE (mins) ↓ |
|---|---|---|---|---|---|---|---|
| | | | | MoCo | | | |
| Retrain | 56.24 | 62.23 | 58.60 | 58.43 | 59.53 | - | 109.47 |
| Fine-Tune | 46.05 (10.19) | 63.49 (1.27) | 58.88 (0.29) | 59.80 (1.37) | 48.81 (10.72) | 4.77 | 1.42 |
| Grad. Ascent | 44.01 (12.23) | 62.66 (0.33) | 59.00 (0.41) | 60.65 (2.22) | 53.28 (6.25) | 3.96 | 0.17 |
| NegGrad | 53.58 (2.66) | 63.70 (1.47) | 58.78 (0.19) | 58.85 (0.42) | 48.68 (10.84) | 3.12 | 1.70 |
| $\ell_1$-Sparsity | 45.68 (10.56) | 60.89 (1.34) | 57.40 (1.20) | 58.66 (0.23) | 52.48 (7.04) | 4.07 | 1.43 |
| AC (Ours) | 50.17 (6.07) | 63.20 (0.97) | 58.56 (0.04) | 58.44 (0.00) | 54.15 (5.38) | **2.49** | 1.87 |
| | | | | SimCLR | | | |
| Retrain | 51.20 | 57.76 | 56.25 | 55.86 | 65.60 | - | 151.77 |
| Fine-Tune | 40.85 (10.35) | 57.29 (0.48) | 54.96 (1.29) | 55.85 (0.00) | 60.61 (4.99) | 3.42 | 1.93 |
| Grad. Ascent | 34.00 (17.21) | 62.12 (4.36) | 59.58 (3.32) | 61.08 (5.23) | 54.21 (11.39) | 8.30 | 0.19 |
| NegGrad | 46.39 (4.81) | 56.52 (1.24) | 54.30 (1.95) | 55.00 (0.86) | 60.04 (5.56) | 2.89 | 2.34 |
| $\ell_1$-Sparsity | 40.55 (10.66) | 57.85 (0.09) | 55.76 (0.49) | 56.68 (0.83) | 58.46 (7.15) | 3.84 | 1.96 |
| AC (Ours) | 46.72 (4.48) | 57.11 (0.65) | 54.70 (2) | 55.23 (0.63) | 59.78 (5.83) | **2.63** | 3.00 |

Table 5: Performance of methods on randomly forgetting 50% of CIFAR-100 training data. EMIA is evaluated on the unlearned encoder, while RA, TA, UA, and MIA are evaluated after linear probing. We report the average gap (Avg. Gap) over these 5 metrics between methods and Retrain. The results are averaged over 5 random trials.

| Methods | EMIA | RA | TA | UA | CMIA | Avg. Gap | RTE |
|---|---|---|---|---|---|---|---|
| | | | | MoCo | | | |
| Retrain | 60.40 | 57.72 | 52.58 | 52.32 | 67.30 | - | 66.71 |
| Fine-Tune | 53.58 (6.81) | 61.9 (4.18) | 56.07 (3.48) | 56.87 (4.55) | 52.27 (15.03) | 6.81 | 0.86 |
| Grad. Ascent | 43.76 (16.63) | 60.91 (3.19) | 56.2 (3.62) | 57.48 (5.16) | 55.64 (11.66) | 8.05 | 0.45 |
| NegGrad | 38.95 (21.44) | 60.94 (3.22) | 57.01 (4.43) | 58.21 (5.89) | 57.64 (9.66) | 8.93 | 1.66 |
| $\ell_1$-Sparsity | 49.97 (10.43) | 58.89 (1.17) | 53.07 (0.49) | 53.66 (1.33) | 56.27 (11.03) | 4.89 | 0.87 |
| AC (Ours) | 56.22 (4.18) | 59.53 (1.81) | 53.37 (0.79) | 53.69 (1.37) | 52.27 (15.03) | **4.63** | 1.84 |
| | | | | SimCLR | | | |
| Retrain | 56.00 | 50.40 | 48.46 | 47.68 | 69.47 | - | 89.74 |
| Fine-Tune | 53.89 (2.12) | 52.82 (2.42) | 49.87 (1.41) | 51.04 (3.36) | 60.06 (9.41) | 3.74 | 1.17 |
| Grad. Ascent | 40.28 (15.72) | 56.32 (5.92) | 54.12 (5.66) | 55.22 (7.54) | 61.43 (8.04) | 8.58 | 0.59 |
| NegGrad | 46.83 (9.17) | 50.60 (0.20) | 48.20 (0.26) | 49.29 (1.62) | 58.01 (11.47) | 4.54 | 2.32 |
| $\ell_1$-Sparsity | 45.12 (10.89) | 52.62 (2.22) | 50.09 (1.62) | 50.98 (3.30) | 65.25 (4.22) | 4.45 | 1.19 |
| AC (Ours) | 54.98 (1.02) | 49.28 (1.12) | 46.8 (1.66) | 47.00 (0.68) | 57.78 (11.69) | **3.24** | 3.07 |

Table 6: Performance of methods on randomly forgetting 10% of SVHN training data from a pre-trained MoCo encoder. The results are averaged over 5 random trials.

| Methods | EMIA | RA | TA | UA | CMIA | Avg. Gap | RTE |
|---|---|---|---|---|---|---|---|
| Retrain | 51.24 | 91.17 | 92.02 | 90.59 | 20.46 | - | 114.13 |
| Fine-Tune | 50.69 (0.54) | 90.19 (0.98) | 90.63 (1.39) | 89.75 (0.84) | 17.03 (3.43) | 1.44 | 1.41 |
| Grad. Ascent | 46.45 (4.79) | 91.23 (0.06) | 92.25 (0.22) | 90.94 (0.35) | 18.44 (2.02) | 1.49 | 0.17 |
| NegGrad | 50.99 (0.24) | 90.67 (0.50) | 91.14 (0.88) | 90.08 (0.50) | 17.42 (3.03) | 1.03 | 1.74 |
| $\ell_1$-Sparsity | 50.56 (0.67) | 89.20 (1.97) | 89.72 (2.31) | 88.89 (1.70) | 20.52 (0.06) | 1.34 | 1.45 |
| AC (Ours) | 51.50 (0.26) | 90.19 (0.98) | 90.76 (1.27) | 89.23 (1.36) | 19.99 (0.46) | **0.87** | 1.98 |

exhibits a more significant unlearning effect under such evaluation. For individual data owners 👤$^i$, the size of their subset $|\mathcal{D}_{\texttt{unlearn}}^i|$ may be small. Therefore, we provide additional qualitative results for visual evaluation: we randomly select 8 samples from $\mathcal{D}_{\texttt{unlearn}}$ to simulate the budget of 👤$^i$. We construct AM (before/after unlearning with $AC$) and AGM for this small set and plot their heatmaps in Figure 3 and observe the apparent

Table 7: Performance of methods on randomly forgetting 10% of MS-COCO data from a pre-trained CLIP. We report the image-text cosine similarity of the retain dataset and unlearn dataset respectively, as well as the average absolute gap from Retrain. The results are averaged over 3 random trials.

| Dataset | Pre-train | Retrain | Fine-Tune | Grad. Ascent | NegGrad | $\ell_1$-Sparsity | AC (Ours) |
|---|---|---|---|---|---|---|---|
| Retrain | 62.09 | 62.47 (0) | 58.67 (3.80) | 61.96 (0.51) | 58.57 (3.90) | 57.70 (4.76) | 60.75 (1.72) |
| Unlearn | 62.08 | 49.84 (0) | 54.75 (4.91) | 62.03 (12.19) | 49.19 (0.65) | 53.54 (3.70) | 51.23 (1.39) |
| Avg. Gap (%) ↓ | - | 0 | 4.35 | 6.35 | 2.28 | 4.23 | **1.56** |

Table 8: Forgetting score (FS) of methods for CIFAR-10 and MoCo. FS gaps are computed between Retrain and other methods.

| Methods | 10% | | 50% | |
|---|---|---|---|---|
| | FS | Gap ↓ | FS | Gap ↓ |
| Retrain | 0.0266 | - | 0.0604 | - |
| Fine-Tune | 0.0393 | 0.0127 | 0.0423 | 0.0180 |
| Grad.Ascent | 0.0005 | 0.0262 | 0.0007 | 0.0596 |
| NegGrad | 0.0205 | 0.0061 | 0.1002 | 0.0398 |
| $\ell_1$-Sparsity | 0.0216 | 0.0050 | 0.0408 | 0.0195 |
| AC (Ours) | 0.0259 | **0.0007** | 0.0672 | **0.0068** |

Figure 2: Negative alignment of 4500 unlearn samples (10%) and MoCo and CIFAR-10. The error bar is the standard deviation.

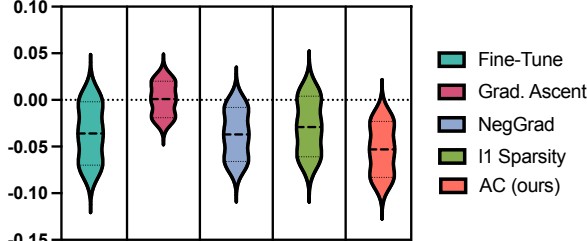

effect of unlearning. We provide additional results for other methods and CLIP unlearning in Figure 6 and Figure 7 in Appendix B.2, where *AC* consistently exhibits the best performance under visual evaluation.

To demonstrate the statistical reliability of negative alignment gap as a black-box evaluation metric, we consider using it to distinguish between the null hypothesis $H_0$ where the model owner replaces the encoder (trained with random seed 11) with another one trained using random seed 10, which still contains information about the unlearned data, and the alternative hypothesis $H_1'$ where unlearning has been performed using AC with $\alpha = \gamma = 1, \beta = 8$, and random seed 11. We compute the average negative alignment gap and its standard deviation across the unlearn data in the task of unlearning 10% data from a MoCo encoder. The resulting statistics are: $H_0 : (\mu_0' = 0.0057, \sigma_0' = 0.0403); H_1' : (\mu_1' = -0.0359, \sigma_1' = 0.0295)$. Note that a data owner possessing $N$ unlearn images can obtain $\frac{N(N-1)}{2}$ negative alignment values. Assuming Gaussian distributions, we perform a t-test to evaluate the $p$-value for distinguishing between $H_0'$ and $H_1'$. We observe from Table 9 that the $p$-value for $N = 5$ is sufficiently small such that the data owner can confidently exclude the null hypothesis $H_0'$. Table 9 also shows that t-test with FS fails in distinguishing AC-unlearning and the cheating case (See Appendix B.9 for more details).

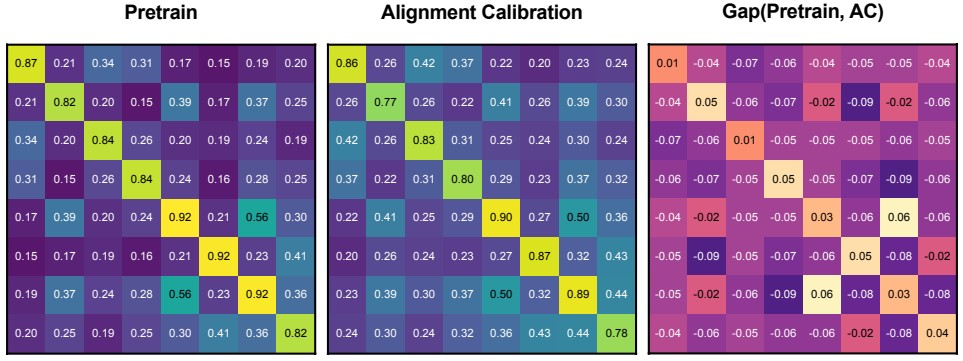

Figure 3: `Alignment Matrices` and `Alignment Gap Matrix` on 8 random images in unlearn dataset of CIFAR-10 (MoCo). The task is forgetting 10% of training data.

Table 9: $p$-value of the t-test on distinguishing between hypothesis $H_0$ and $H_1'$.

| Metric $\backslash N$ | 5 | 10 | 15 | 20 |
|---|---|---|---|---|
| FS | 0.7452 | 0.64 | 0.5648 | 0.5052 |
| Neg. Align. Gap | **0.0168** | **≤0.0001** | **≤0.0001** | **≤0.0001** |

Table 10: Ablation study on positive and negative calibration in Equation (5) regarding the average gap over metrics on CIFAR-10 and MoCo with forgetting ratio 10/50%. "w/" denote with and "w/o" denotes without. For example, "w/o, w/" means $AC$ without negative calibration but with positive calibration.

| Neg. Cal. | Pos. Cal. | Forgetting 10% | 50% |
|---|---|---|---|
| w/ | w/ | 0.92 | 2.00 |
| w/o | w/ | 1.25 | 4.12 |
| w/ | w/o | 1.95 | 3.75 |
| w/o | w/o | 2.60 | 6.45 |

Figure 4: The effect of $\alpha$ and $\beta$ on the forgetting score ratio between Retrain and $AC$, *i.e.*, FS(RT):FS(AC). Here we forget 10% of CIFAR-10 training data from a MoCo encoder.

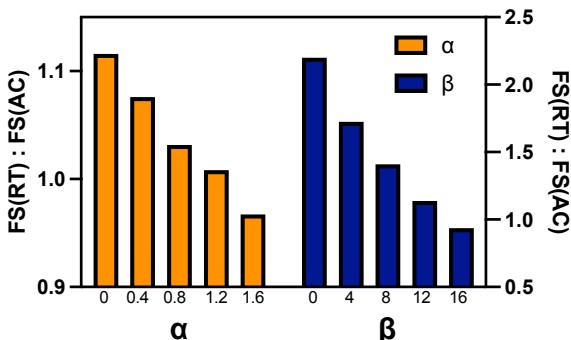

## 5.4 Ablation Study

**Influence of negative alignment calibration:** In Equation (5), the coefficient $\alpha$ controls the intensity of maximizing the negative alignment on unlearn data. To explore the effect of negative alignment calibration in the unlearning task, we fix $\beta$ and adjust $\alpha$ while keeping $\gamma = \alpha$ for simplicity. Figure 4 (orange bars) reports the ratio between the forgetting score FS of Retrain and $AC$. When $\alpha$ increases, the ratio decreases, indicating that the resulting model forgets more information about the unlearn data. A ratio of 1 denotes that the FS of $AC$ equals that of Retrain. Furthermore, we consider a more extreme case of $\alpha = 0$, representing no negative calibration. In Table 10, without negative calibration, the average gap over metrics is larger than that of the standard $AC$ by 0.33/2.12% (comparing rows "w/o, w/" with "w/, w/") for 10/50% forgetting tasks, suggesting this additional term not only benefits the data owner for unlearn evaluation but also improves the unlearn performance.

**Influence of positive alignment calibration.** In Equation (5), the coefficient $\beta$ controls the intensity of minimizing the positive alignment on the unlearn data. In Figure 4 (blue columns), we fix $\alpha = \gamma = 1$ and vary $\beta$ from 0 to 16. The forgetting score ratio decreases with increasing $\beta$ and approximately reaches 1.0 in the range of [12,16]. In Table 10, the positive alignment calibration term enhances the unlearning performance from 1.95/3.75% to 0.92/2% (comparing columns "w/, w/o" with "w/, w/") for the 10/50% forgetting tasks regarding the average gap.

## 5.5 Preserving Uniformity

Finally, we validate the function of the performance preserving term in our loss function, which forces extracted features to spread uniformly on the hyper-sphere. Following the implementation from Wang & Isola (2020), we visualize the uniformity of features on CIFAR-10 before and after applying our AC unlearning algorithm. Specifically, we train a ResNet-18 encoder mapping images to 2D space using SimCLR and plot the distribution of angles (*i.e.*, $\arctan2(x, y)$ for each point $(x, y) \in \mathcal{S}^1$) in Figure 5. We observe that this term implicitly maintains the uniformity of the representation on both the test dataset and the unlearn dataset.

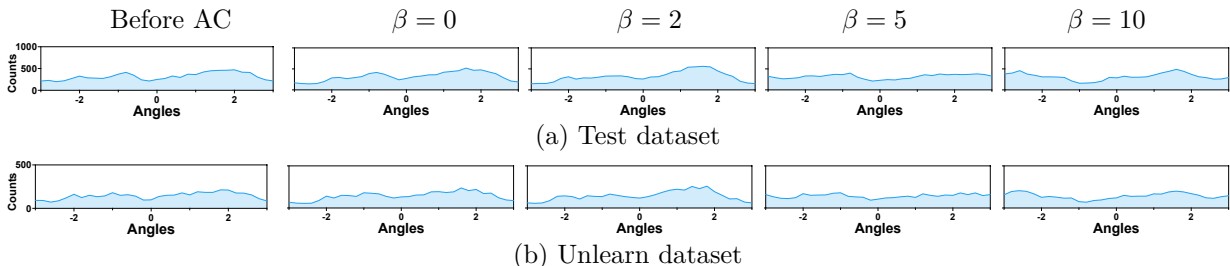

Figure 5: Angle distributions on **test dataset** and **unlearn dataset** before and after AC unlearning with different $\beta$ values related to positive alignment calibration intensity. Here, the negative alignment calibration and performance-preserving term intensities are kept at their default values, *i.e.*, $\alpha = \gamma = 1$. Note that when $\beta = 0$, only negative alignment calibration and performance-preserving terms take effect in our AC method. Each chart's x-axis represents "Angles" and the y-axis represents "Counts".

## 6  Conclusion

In this paper, we study the problem of machine unlearning for contrastive learning pre-training (MUC). We establish the foundations on this line of study by adapting existing unlearning methods and setting up baseline evaluation metrics, including *white-box evaluation* for model owners to choose an optimal unlearning strategy, and *black-box auditing* for data owners to examine the effect of unlearning. After identifying the suboptimality of existing unlearning methods and the insufficiency of current auditing tools, we further propose our novel method called *Alignment Calibration*. Our approach introduces a novel unlearning objective function to strategically optimize toward the unlearning goal and enable straightforward visual auditing. Empirically, our method achieves state-of-the-art performance on unlearning tasks for both unimodal and multimodal contrastive learning.

**Limitations and Future Work:**  (1) Our paper initializes the study of machine unlearning in self-supervised learning but only considers contrastive learning. We plan to extend our exploration of unlearning towards other SSL methods in the future. (2) Pseudo-label-based (Warnecke et al., 2021) and label-free (Foster et al., 2024a;b) unlearning algorithms may be integrated into our MUC framework as future enhancements. (3) Our paper focuses on approximate unlearning methods for contrastive learning and does not provide theoretical guarantees (e.g., certification of removal), which represents an interesting direction for future research. (4) While our work focus on random forgetting, study the difficulty of removing specific samples is important in the contrastive learning setting. We plan to extend existing studies (e.g., (Zhao et al., 2024)) to contrastive learning in future work. (5) Finally, although we perform t-test to show the advantages of our proposed evaluation tool, *i.e.*, the negative alignment values and `AGM`, we consider one typical deceptive case for the null hypothesis where the model owner cheats by replacing the encoder with a different one without unlearning. In future work, we will investigate more failure cases where malicious model owners deliberately design cheating methods to deceive the metrics proposed in this paper, and propose mitigation strategies accordingly.

## Acknowledgement

YY gratefully acknowledges funding support from NSERC, the Ontario early researcher program and the Canada CIFAR AI Chairs program. Resources used in preparing this research were provided, in part, by the Province of Ontario, the Government of Canada through CIFAR, and companies sponsoring the Vector Institute. XS gratefully acknowledges funding support by the Strategic Priority Research Program of CAS Grant XDA0480502, Robotic AI-Scientist Platform of CAS, and NSFC Grant 12288201. This research was also funded by the Deutsche Forschungsgemeinschaft (DFG, German Research Foundation), Project number 550224287.

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

# A Experiment Details

## A.1 Datasets

**CIFAR-10/100.** Both datasets consist of 50K training images and 10K test images. All the images are 32x32 colored. CIFAR-100 has 100 categories and CIFAR-10 has 10 categories. In unimodal contrastive learning, the augmentations for training encoders include random resizing and cropping, random grayscale, random color jitter, and horizontal flipping. We split the 50K training images into a validation set of 5K images and a training set of 45K images. For example, when the unlearning task is to forget 10% of training data, the unlearn dataset $\mathcal{D}_{\texttt{unlearn}}$ has 4.5K images and the retain dataset $\mathcal{D}_{\texttt{retain}}$ has 4.05K images.

**SVHN** SVHN consists of 73,257 training images and 26,032 test images of 10 categories. All the images are 32x32 colored. We split the 10% of training images into a validation set and the rest into a training set.

**MS-COCO.** COCO is a large-scale object detection, segmentation, and captioning dataset. Its training set contains 118,287 images and 591,753 captions. Each image has several objects and corresponds to at least 5 captions. Different from unimodal contrastive learning which uses strong augmentations, CLIP employs only resizing, center cropping and horizontal flipping to make images of 224x224 pixels.

## A.2 Contrastive Learning and Unlearning Methods

**MoCo and SimCLR:** For the pre-trained (clean) models, we train the encoder for 800 epochs using an SGD optimizer with cosine-scheduled learning rate initialized at 0.06, momentum of 0.9, and weight decay of 0.0005. For unlearning methods: *Retrain* applies the same training strategy as pre-training; *Fine-tuning* and *NegGrad* updates the pre-trained encoder for 10 epochs with a learning rate searched in [0.003, 0.03]; *Gradient Ascent* updates the pre-trained encoder using reversed stochastic gradient descent for 5 epochs with a learning rate searched in $[10^{-6}, 10^{-4}$; $\ell_1$-*Sparsity* applies the learning rate as 0.006 and implements $\ell_1$ regularization with a coefficient searched in $[10^{-6}, 10^{-3}]$. For our *Alignment Calibration* method, we update the pre-trained encoder for 10 epochs and search the learning rate in [0.003, 0.03] and the tunable parameter $\beta$ in [0, 20] for different unlearning tasks. If not otherwise stated, we adopt $\alpha = \gamma = 1$. For simplicity, in our reported results on CIFAR-10/100, we use a learning rate of 0.006 for 10% forgetting, and 0.02 for 50% forgetting. The linear probing stage trains a linear classifier head for 100 epochs using an SGD optimizer with a cosine-scheduled learning rate initialized at 1.0, and a momentum of 0.9. The batch size is set as 512 for both encoder and linear head training.

**CLIP:** For the pre-trained (clean) CLIP, we train the model for 35 epochs on 2 NVIDIA RTX 4090 GPUs using an AdamW optimizer with a warm-up cosine-scheduled learning rate initialized at 5e-4 and momentum of 0.9. The total batch size is 256 (128 on each GPU). For unlearning algorithms: *Retrain* again applies the same training strategy as pre-training; *Fine-tuning* update the pre-trained model for 8 epochs with a fixed learning rate searched in [5e-5,5e-4]; *NegGrad* updates the pre-trained model for 8 epochs with a fixed learning rate searched in $[10^{-5}, 10^{-4}]$; *Gradient Ascent* updates the pre-trained model for 4 epochs with a fixed learning rate searched in [5e-6, 5e-4]; $\ell_1$-*Sparsity* updates the pre-trained model for 8 epochs with a learning rate of 0.0005 and a regularization coefficient searched in $[10^{-9}, 10^{-4}]$. For our *Alignment Calibration* method, we update the pre-trained model for 8 epochs with a fixed learning rate of 0.0002. We search $\alpha = \gamma$ in [0.5, 1] and $\beta$ in [0, 1].

## A.3 Unlearning Evaluation

**CMIA efficacy.** Given an unlearned encoder $\hat{g}$, we execute linear probing on it and denote the whole classifier by $f$. Following the implementation of Jia et al. (2023); Fan et al. (2024), we evaluate the attack successful rate (ASR) on the unlearn dataset $\mathcal{D}_{\texttt{unlearn}}$ of a confidence-based membership inference attack (Song & Mittal, 2021) to $f$. The formal definition of CMIA efficacy is given by:

$$\texttt{CMIA-Efficacy} := \frac{TN_{\texttt{CMIA}}}{|\mathcal{D}_{\texttt{unlearn}}|}, \tag{6}$$

where $TN_{\texttt{CMIA}}$ is the number of true negatives predicted by the CMIA attack.

**EMIA efficacy.** We implement the alignment-based EncoderMI-T attack (Liu et al., 2021) in an adapted white-box setting. Given an unlearned encoder $\hat{g}$ with its retain dataset $\mathcal{D}_{\texttt{retain}}$ and test dataset $\mathcal{D}_{\texttt{test}}$, we denote $\mathcal{D}_{\texttt{non-member}} \coloneqq \mathcal{D}_{\texttt{test}}$ sample a subset $\mathcal{D}_{\texttt{member}}$ of $\mathcal{D}_{\texttt{retain}}$ such that $|\mathcal{D}_{\texttt{non-member}}| = |\mathcal{D}_{\texttt{member}}|$. For each data in $\mathcal{D}_{\texttt{non-member}}$ and $\mathcal{D}_{\texttt{member}}$, we first augment it 10 times and compute features of these 10 views via $\hat{g}$. Then, we compute the cosine similarity between each pair of features, *i.e.*, 45 pairs, and take the average of these similarity values. Now we get a membership feature dataset and a non-membership feature dataset whose data points are just scalar values. The EncoderMI-T attack then searches for an optimal threshold to classify membership features and non-membership features. Similar to MIA efficacy, the formal definition of EMIA efficacy is given by:

$$\texttt{EMIA-Efficacy} \coloneqq \frac{TN_{\texttt{EMIA}}}{|\mathcal{D}_{\texttt{unlearn}}|}, \tag{7}$$

where $TN_{\texttt{EMIA}}$ is the number of true negatives predicted by the EncoderMI attack.

## B  Additional Experiments

### B.1  Unlearning Performance for More Tasks

We present experiment results on CIFAR-100 in Tables 4 and 5. Across these different tasks, our proposed *Alignment Calibration* method achieves the lowest average gap compared to the Retrain method.

### B.2  More Visual Visualization Results

In Figure 6, we report the $\texttt{AGM}$ of Retrain, Fine-Tune, Gradient Ascent, NegGrad, $\ell_1$-Sparsity on CIFAR-10, as a complement to Figure 3. For the unlearning task on CLIP, we check the $\texttt{ASM}$ of our *AC* and other baseline methods in Figure 7.

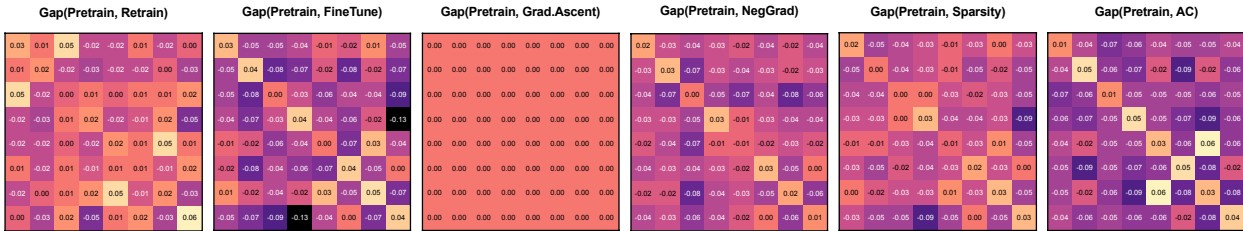

Figure 6: $\texttt{Alignment Gap Matrices}$ of 8 unlearn images for Retrain, Fine-Tune, Gradient Ascent, NegGrad, $\ell_1$-Sparsity, and our *Alignment Calibration*. The unlearning task is to forget 10% of CIFAR-10 training data from a MoCo encoder.

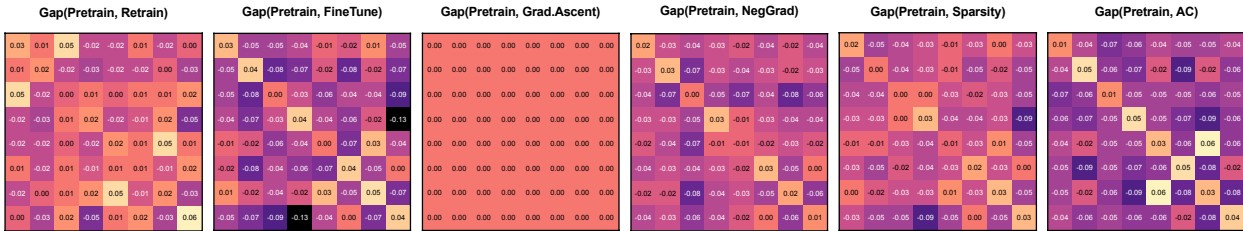

Figure 7: $\texttt{Alignment Gap Matrices}$ of 6 unlearn image-text pairs for Retrain, Fine-Tune (FT), Gradient Ascent (GA), NegGrad (NG), $\ell_1$-Sparsity, and *Alignment Calibration (AC)*. The unlearning task is to forget 10% of MS-COCO training data from a CLIP encoder.

### B.3 Comparison using Average Gap Percentage Standard

In previous experiments, we compare our AC and baseline methods using the average gap across multiple metrics. To comprehensively illustrate the advantage of our method, we introduce a different standard for comparison, *i.e.*, average gap percentage (AGP), which averages the percentage of GAP over Retrain across multiple metrics. In Table 11, we perform unlearning tasks of forgetting 10%/50% of CIFAR-10 data from ResNet-18 SimCLR/MoCo encoders and our proposed AC method still outperforms baseline methods concerning the AGP standard.

Table 11: Comparison in average gap percentage (AGP, %) with other baseline methods. AGP averages the percentages of Gap over Retrain, *i.e.*, MEAN($\{\frac{gap}{retrain}\}$).

| CIFAR-10 | 10% | | 50% | |
|---|---|---|---|---|
| | MoCo | SimCLR | MoCo | SimCLR |
| Fine-Tune | 3.81 | 5.23 | 9.85 | 9.01 |
| Grad. Ascent | 5.83 | 8.19 | 14.05 | 13.35 |
| NegGrad | 3.91 | 6.09 | 5.40 | 9.10 |
| $\ell_1$-Sparsity | 3.22 | 4.70 | 7.46 | 10.31 |
| AC (ours) | **2.16** | **3.61** | **4.07** | **7.75** |

Table 12: Unlearning performance of randomly forgetting 10 % of CIFAR-10 training data from ResNet-50 MoCo encoders. We compare AC with baseline methods in Average Gap (AG, %) and the Average Gap Percentage (AGP, %)

| | EMIA | RA | TA | UA | CMIA | AG | AGP |
|---|---|---|---|---|---|---|---|
| Retrain | 53.56 | 92.10 | 90.10 | 90.98 | 35.44 | - | - |
| Fine-Tune | 49.85 | 91.74 | 89.64 | 91.04 | 23.94 | 2.68 | 8.07 |
| Grad. Ascent | 39.36 | 92.36 | 90.71 | 92.32 | 25.35 | 4.43 | 11.49 |
| NegGrad | 52.62 | 91.85 | 89.78 | 90.64 | 24.88 | 2.07 | 6.51 |
| $\ell_1$-Sparsity | 47.61 | 90.54 | 88.67 | 90.19 | 27.04 | 3.63 | 7.79 |
| AC (ours) | 53.48 | 91.75 | 89.32 | 89.82 | 30.56 | **1.45** | **3.29** |

### B.4 Model Architecture

In Table 12, we perform an unlearning task of forgetting 10% of CIFAR-10 data from a ResNet-50 MoCo encoder. Our AC method outperforms baseline methods concerning both average gap and average gap percentage standards.

Table 13: Standard deviation for Tables 2 to 4. Appendix B.5 shows the detailed settings for random trials.

| | EMIA | RA | TA | UA | CMIA | EMIA | RA | TA | UA | CMIA |
|---|---|---|---|---|---|---|---|---|---|---|
| | CIFAR-10, 10%, MoCo | | | | | CIFAR-10, 50%, MoCo | | | | |
| Retrain | 49.72±5.58 | 89.54±0.07 | 87.76±0.25 | 88.42±0.66 | 34.38±0.83 | 55.95±1.58 | 85.98±0.25 | 83.55±0.27 | 83.98±0.21 | 46.66±0.73 |
| Fine-Tune | 50.15±7.99 | 88.34±0.27 | 86.46±0.13 | 87.59±0.18 | 29.42±1.37 | 49.98±2.92 | 87.89±0.55 | 85.66±0.43 | 86.65±0.29 | 32.35±0.90 |
| Grad. Ascent | 44.95±6.61 | 89.92±0.16 | 88.28±0.23 | 89.76±0.46 | 28.53±0.76 | 42.90±5.46 | 89.51±0.30 | 87.79±0.24 | 88.99±0.17 | 31.85±0.61 |
| NegGrad | 48.43±5.42 | 89.25±0.24 | 87.35±0.23 | 88.58±0.49 | 28.89±0.86 | 57.40±13.24 | 83.04±0.47 | 80.15±0.71 | 80.59±0.56 | 40.65±3.02 |
| $\ell_1$-Sparsity | 49.38±7.15 | 88.56±0.32 | 86.91±0.40 | 88.12±0.39 | 29.91±0.92 | 52.19±16.71 | 81.60±0.76 | 80.19±0.91 | 80.77±0.81 | 38.43±6.48 |
| AC (Ours) | 50.28±5.86 | 89.14±0.19 | 87.24±0.25 | 88.20±0.50 | 31.50±1.17 | 55.02±5.14 | 86.28±0.35 | 83.28±0.37 | 83.72±0.26 | 38.39±1.27 |
| | CIFAR-10, 10%, SimCLR | | | | | CIFAR-10, 50%, SimCLR | | | | |
| Retrain | 48.11±3.70 | 90.87±0.08 | 88.94±0.16 | 89.68±0.33 | 38.87±1.33 | 53.37±2.21 | 87.23±0.25 | 85.16±0.34 | 85.69±0.17 | 49.30±0.65 |
| Fine-Tune | 47.72±4.38 | 89.38±0.51 | 87.26±0.65 | 88.93±0.56 | 30.71±1.61 | 45.90±8.25 | 87.88±0.34 | 85.50±0.34 | 87.23±0.30 | 35.44±1.50 |
| Grad. Ascent | 41.48±1.51 | 91.26±0.13 | 89.55±0.37 | 91.11±0.36 | 29.36±0.99 | 42.23±4.50 | 90.52±0.28 | 88.61±0.15 | 90.45±0.16 | 33.28±2.47 |
| NegGrad | 49.56±13.95 | 89.10±0.32 | 87.23±0.45 | 89.07±0.43 | 29.97±1.13 | 55.70±13.70 | 83.98±0.90 | 82.15±0.68 | 83.80±0.55 | 33.67±5.75 |
| $\ell_1$-Sparsity | 48.44±6.70 | 90.59±0.23 | 88.56±0.30 | 90.44±0.28 | 30.61±1.01 | 46.51±3.85 | 89.84±0.12 | 87.75±0.28 | 89.48±0.22 | 35.38±1.52 |
| AC (Ours) | 48.64±9.47 | 90.24±0.11 | 88.06±0.23 | 89.24±0.36 | 33.12±1.54 | 47.12±5.84 | 86.11±0.31 | 83.92±0.30 | 85.24±0.60 | 37.57±1.16 |
| | CIFAR-100, 10%, MoCo | | | | | CIFAR-100, 10%, SimCLR | | | | |
| Retrain | 56.24±2.77 | 62.23±0.27 | 58.60±0.28 | 58.43±0.64 | 59.53±1.57 | 51.20±2.07 | 57.76±0.30 | 56.25±0.31 | 55.86±0.78 | 65.60±1.70 |
| Fine-Tune | 46.05±4.96 | 63.49±0.56 | 58.88±0.44 | 59.80±0.91 | 48.81±1.21 | 40.85±3.81 | 57.29±0.13 | 54.96±0.43 | 55.85±1.04 | 60.61±3.70 |
| Grad. Ascent | 44.01±2.70 | 62.56±0.11 | 59.00±0.06 | 60.65±0.64 | 53.28±2.12 | 34.00±3.81 | 62.12±0.25 | 59.58±0.22 | 61.08±0.61 | 54.21±1.91 |
| NegGrad | 53.58±4.82 | 63.70±0.25 | 58.78±0.27 | 58.85±0.53 | 48.68±2.02 | 46.39±5.10 | 56.52±0.52 | 54.30±0.67 | 55.00±1.08 | 60.04±3.13 |
| $\ell_1$-Sparsity | 45.68±7.30 | 60.89±0.41 | 57.40±0.59 | 58.66±0.78 | 52.48±3.21 | 40.55±3.43 | 57.85±0.24 | 55.76±0.34 | 56.68±0.64 | 58.46±5.10 |
| AC (Ours) | 50.17±3.42 | 63.20±0.37 | 58.56±0.59 | 58.44±0.42 | 54.15±1.66 | 46.72±5.50 | 57.11±0.38 | 54.70±0.36 | 55.23±0.91 | 59.78±5.46 |
| | CIFAR-100, 50%, MoCo | | | | | CIFAR-100, 50%, SimCLR | | | | |
| Fine-Tune | 53.58±3.92 | 61.90±0.56 | 56.07±0.44 | 56.87±0.41 | 52.27±0.62 | 53.89±9.87 | 52.82±0.29 | 49.87±0.46 | 51.04±0.38 | 60.06±2.23 |
| Grad. Ascent | 43.76±3.62 | 60.91±0.14 | 56.20±0.18 | 57.48±0.31 | 55.34±0.40 | 40.28±5.87 | 56.32±0.40 | 54.12±0.75 | 55.22±0.88 | 61.43±2.03 |
| NegGrad | 38.95±4.97 | 60.94±0.47 | 57.01±0.37 | 58.21±0.35 | 57.64±1.01 | 46.83±7.95 | 50.60±0.37 | 48.20±0.45 | 49.29±0.65 | 58.01±1.90 |
| $l_1$-Sparsity | 49.97±4.97 | 58.89±0.47 | 53.07±0.37 | 53.66±0.35 | 56.27±1.01 | 45.12±5.09 | 52.62±0.63 | 50.09±0.79 | 50.98±0.56 | 65.25±2.31 |
| AC (Ours) | 56.22±5.32 | 59.53±0.78 | 53.37±0.75 | 53.69±0.32 | 52.27±4.08 | 54.98±2.85 | 49.28±0.57 | 46.80±0.61 | 47.00±0.50 | 57.78±4.66 |

### B.5 Random Trial Settings

Recall that our results in Tables 2 to 4 are averaged results over five random trials for each unlearning setting. Specifically, we use random seeds 0, 1, 2, 3, and 4. Each random seed determines both the selection of unlearning samples and the corresponding pre-trained and retrained encoders. For instance, with a specific

Table 14: Average gaps calculated only cross classifier-level metrics, *i.e.*, RA, TA, UA, and CMIA. The hyper-parameters for unlearning methods are chosen optimally regarding this criterion. **Bold** font denotes the lowest average gap and *italic* font denotes the second lowest. The average standard deviation across the four metrics is shown in square brackets.

| | CIFAR-10 | | | | CIFAR-100 | | | |
| | 10% | | 50% | | 10% | | 50% | |
| | MoCo | SimCLR | MoCo | SimCLR | MoCo | SimCLR | MoCo | SimCLR |
| --- | --- | --- | --- | --- | --- | --- | --- | --- |
| FineTune | 1.69 [0.52] | *2.37* [0.57] | 5.25 [0.54] | **3.53** [0.59] | 2.85 [0.73] | 1.69 [0.75] | 6.44 [0.31] | 3.64 [0.83] |
| Grad. Ascent | 1.92 [0.39] | 2.98 [0.46] | 6.90 [0.32] | 6.88 [0.77] | *2.30* [0.89] | 6.07 [1.33] | 5.80 [0.47] | 6.79 [1.01] |
| NegGrad | *1.59* [0.46] | 2.79 [0.41] | *3.58* [0.52] | 4.64 [0.87] | 3.00 [0.90] | 2.14 [1.62] | **3.51** [0.99] | 3.39 [0.84] |
| $\ell_1$-Sparsity | 1.65 [0.51] | 2.39 [0.31] | 4.80 [2.24] | 4.76 [0.74] | 2.31 [1.01] | *2.07* [1.56] | 5.90 [0.55] | **2.84** [1.07] |
| AC (Ours) | **0.91** [0.46] | **1.89** [0.28] | **2.20** [0.45] | *3.64* [0.59] | **1.60** [0.76] | **1.54** [1.46] | *4.44* [0.59] | *3.37* [1.37] |

random seed, our AC method and other baseline methods start from the same pre-trained encoder to forget the same samples and aim to approximate the same retrained encoder. The metrics are computed in each random trial. We report the standard derivation for Tables 2 to 4 in Table 13.

We observe that EMIA results demonstrate significant instability with respect to random initialization, exhibiting large variances across all baseline methods as well as our proposed approach. To address potential concerns regarding the reliability of this evaluation metric, we conduct additional assessments using only classifier-level metrics (RA, TA, UA, and CMIA). For this evaluation, we carefully select hyperparameters for all unlearning methods according to these criteria to ensure optimal performance and report the results in Table 14. The results consistently show that our method still achieves either first or second-best performance across all tasks, while maintaining low standard deviation. This substantiates the robustness of our approach even when evaluated through alternative, potentially more reliable metrics.

## B.6 Time and Memory Consumption

We acknowledge the running time trade-off introduced by AC. To further examine the practicality of AC, we provide additional results in Table 15, which provides detailed runtime and GPU memory usage across models ranging from ResNet-18 to ResNet-152. Our results demonstrate that AC introduces minimal computational overhead even for the largest models tested.

Table 15: Comparison of time and GPU memory consumption across networks of different sizes. Experiments are conducted on a server with four NVIDIA A100 GPUs. GPU Memory is monitored by the Wandb logger.

| | Time (minutes) | | | | | GPU Memory (GB) | | | | |
| | RN-18 | RN-34 | RN-50 | RN-101 | RN-152 | RN-18 | RN-34 | RN-50 | RN-101 | RN-152 |
| --- | --- | --- | --- | --- | --- | --- | --- | --- | --- | --- |
| Retrain | 78.09 | 127.44 | 243.95 | 384.44 | 543.46 | 5.19 | 6.91 | 17.89 | 24.56 | 32.62 |
| Fine-Tune | 1.03 | 1.62 | 3.11 | 4.89 | 6.90 | 5.19 | 6.91 | 17.89 | 24.56 | 32.62 |
| Grad. Ascent | 0.14 | 0.14 | 0.25 | 0.36 | 0.47 | 5.19 | 6.91 | 17.89 | 24.56 | 32.62 |
| NegGrad | 1.22 | 1.96 | 3.77 | 5.92 | 8.33 | 5.19 | 6.91 | 17.89 | 24.56 | 32.62 |
| $\ell_1$-Sparsity | 1.04 | 1.65 | 3.15 | 4.91 | 6.99 | 5.19 | 6.91 | 17.89 | 24.56 | 32.63 |
| AC (Ours) | 1.33 | 2.18 | 4.16 | 6.55 | 9.19 | 5.21 | 6.93 | 17.91 | 24.58 | 32.62 |

## B.7 Privacy concerns

We acknowledge that from a differential privacy perspective, visible changes from AC could theoretically enable membership inference attacks by observing differences in model behavior compared to retraining (Sekhari et al., 2021), as validated in our AGM analysis in Figure 8. However, we argue that such privacy leakage poses minimal practical risk because:

- *Limited data access*: AGM calculations require access to the private unlearn data, which is typically only available to data owners and model developers;

- *No baseline availability*: In practice, the retraining process is not performed and retrained models are not released, eliminating the comparison baseline needed for inference;

- *Restricted adversary capabilities*: An adversary would need simultaneous access to both the unlearned model and reference data/models, which is unlikely in real deployment scenarios

In summary, while the theoretical privacy concern is valid, the practical barriers to exploitation make such risks negligible in realistic threat models where adversaries lack access to the necessary private information for meaningful membership inference.

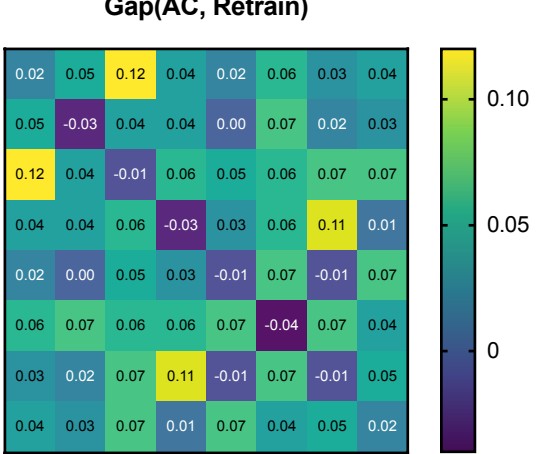

Figure 8: The Gap metric between Retrain and AC on 8 random images in unlearn dataset of CIFAR-10 (MoCo). The task is forgetting 10% of training data.

### B.8 More Ablation Studies

Understanding the trade-off between forgetting efficacy and utility preservation is crucial for practical unlearning applications. Figure 9 complements Figure 4 by showing how tuning $\alpha$ and $\beta$ affects utility metrics. This provides a complete picture of the forgetting-utility trade-off landscape across different hyperparameter settings.

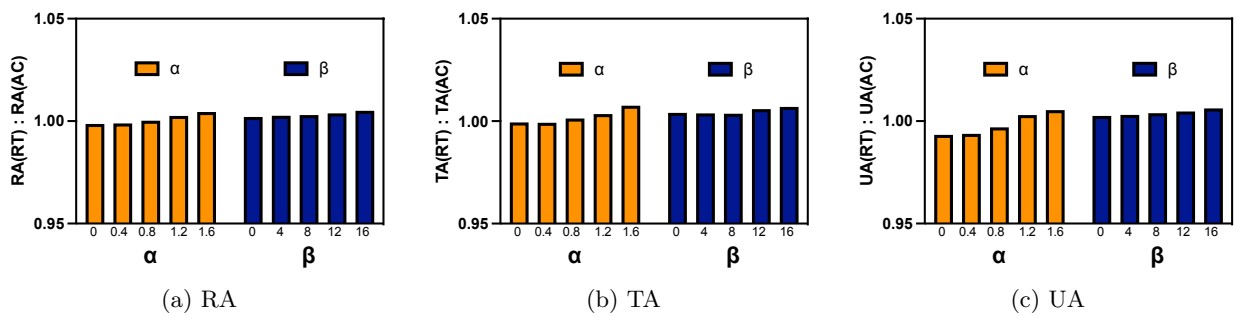

Figure 9: The effect of $\alpha$ and $\beta$ on RA, TA, and UA between Retrain and $AC$, *i.e.*, FS(RT):FS(AC). Here we forget 10% of CIFAR-10 training data from a MoCo encoder.

### B.9  More Details on t-Test

Here we clarify the detailed statistics for the t-test with `FS` in Section 5.3: the null hypothesis (cheating case) $H_0$ has the same mean and standard deviation ($\mu_0 = -0.0026, \sigma_0 = 0.0587$) as shown in Section 3.4, and the alternative hypothesis (AC-unlearnig case) $H_1'$ has ($\mu_1'' = 0.0084, \sigma_1'' = 0.04438$).

