# OpenReview forum: "MUC: Machine Unlearning for Contrastive Learning with Black-box Evaluation"
_TMLR — Accepted by TMLR_

### Review · Reviewer_HyuW · 2025-05-04

**Summary Of Contributions:**

This submission addresses the problem of Machine Unlearning for Contrastive Learning (MUC), focusing on efficiently removing the influence of specific data points (Dunlearn) from pre-trained contrastive models (like SimCLR, MoCo, CLIP) without costly retraining from scratch on the remaining data (Dretain).

The work clearly defines the MUC setting, distinguishing between the needs and capabilities of model owners (white-box evaluation) and data owners (black-box evaluation). It systematically adapts existing supervised unlearning methods (Fine-tuning, Gradient Ascent, NegGrad, $\ell_1$-Sparsity) to the contrastive learning paradigm (using InfoNCE loss) and establishes their performance as baselines, demonstrating their sub-optimality compared to exact retraining.

**Audience:**

Yes

**Claims And Evidence:**

Yes

**Requested Changes:**

Discuss Scalability in More Detail: Add a paragraph discussing potential scalability challenges and computational/memory overhead compared to baselines, especially for very large models or datasets. If possible, provide more detailed profiling beyond RTE (e.g., wall-clock time comparison to Fine-Tune).

**Strengths And Weaknesses:**

Timeliness and Importance: Addresses the critical and increasingly relevant problem of machine unlearning, specifically within the popular and powerful framework of contrastive self-supervised learning.

Clear Problem Definition: Effectively delineates the MUC problem, considering the different perspectives and constraints of model owners and data owners (white-box vs. black-box).

Novel and Well-Motivated Algorithm (AC): The proposed Alignment Calibration method is novel and its objective function components are well-justified, directly targeting the goals of forgetting specific data, retaining general performance, and improving evaluability. The concepts of positive/negative alignment calibration are intuitive.

## weakness
Hyperparameter Sensitivity: The AC objective introduces several hyperparameters ($\alpha, \beta, \gamma, \epsilon$). While some analysis is presented (Figure 4, Table 8), a more thorough sensitivity analysis across different datasets/models regarding their impact on the trade-off between forgetting metrics (FS, MIAs) and utility metrics (RA, TA) would strengthen the work. How crucial is precise tuning for achieving good results?

Scalability Concerns: While efficiency (RTE) relative to retraining is shown, the absolute computational cost and memory requirements, especially compared to lightweight methods like Fine-tuning or Grad Ascent, could be discussed further. How does AC scale to significantly larger models (beyond ResNet-50 mentioned in Appendix) or datasets?

Theoretical Justification: The paper provides strong empirical results and intuition, but lacks deeper theoretical analysis. For instance, are there any theoretical insights or bounds on how well AC approximates exact unlearning in terms of information removal or distribution shift compared to retraining?

---

> ### Author Response · Authors · 2025-05-27
>
> We are grateful for the reviewer's careful evaluation and helpful suggestions, which have enhanced the quality of our submission. We address each of your points below and remain open to further discussion:
>
> [**W1. Hyperparameter Sensitivity**] We appreciate the reviewer's concern about hyperparameter sensitivity and would like to clarify our approach:
>
> - While we introduce four hyperparameters for mathematical generality, only two require tuning in practice. Specifically, $\gamma=1$ follows standard contrastive learning conventions for uniformity preservation, and $\varepsilon= |D_{\texttt{unlearn}}|/|D_{\texttt{retain}}|$ is determined by the dataset sizes rather than being a free parameter. This reduces the hyperparameter space significantly.
> - Importantly, AC achieves state-of-the-art results with minimal tuning—simply setting $\alpha=\gamma=1$ and using the dataset-determined $\varepsilon$ delivers competitive performance across all our experiments. This demonstrates that AC is not critically dependent on extensive hyperparameter optimization.
> - While our default settings achieve strong results, we acknowledge that task-specific tuning could yield additional improvements, which is consistent with standard practice in machine learning.
> - We agree that understanding the trade-off between forgetting efficacy and utility preservation is crucial for practical unlearning applications. We have added Figure 9 in Appendix B.8, which complements Figure 4 by showing how tuning $\alpha$ and $\beta$ affects utility metrics. This provides a complete picture of the forgetting-utility trade-off landscape across different hyperparameter settings.
>
>
>
>
> [**W2: Scalability Concern**]
> We appreciate the reviewer's important question about scalability and have conducted additional experiments to address this concern comprehensively.
> - AC's computational complexity scales linearly with the model size, identical to standard training procedures. The additional operations do not fundamentally alter the asymptotic complexity, making AC inherently scalable to larger architectures.
> - We have added Table 13 in Appendix B.6, which provides detailed runtime and GPU memory usage across models ranging from ResNet-18 to ResNet-152. Our results demonstrate that AC introduces minimal computational overhead even for the largest models tested.
>
> [**W3: Theoretical Analysis**] Providing formal theoretical guarantees for approximate unlearning in deep learning settings remains an open challenge in the field. Most existing theoretical work with certified guarantees (e.g., Differential Privacy (DP)-based methods [1]) is restricted to linear models and often comes with significant performance trade-offs that limit practical applicability.
>
> We agree that developing certified approximate unlearning for contrastive learning represents an important research direction. Our empirical framework and method design provide a foundation for future theoretical analysis.
>
> [1] Chien et al., Certified Machine Unlearning via Noisy Stochastic Gradient Descent, NeurIPS 2024

---

### Review · Reviewer_ownh · 2025-05-10

**Summary Of Contributions:**

The paper introduces addresses the problem of unlearning for models trained with contrastive learning. First, a novel unlearning algorithm (AC) is introduced, which is an extension of the InfoNCE loss. Second, Forgetting Score is adapted to the contrastive setting, and also introduces the notion of an Alignment Matrix and Alignment Gap Matrix, to evaluate the divergence in alignment between augmented outputs before/after unlearning.

The proposed method achieves state of the art performance.

**Audience:**

Yes

**Broader Impact Concerns:**

No concerns for broader impact at this time.

**Claims And Evidence:**

Yes

**Requested Changes:**

Please see the weaknesses section.  I would specifically like to have a discussion on weakness [1], and it would be good to see some updates to literature and/or comparison methods.

One additional minor correction to the abstract: Contrastive learning methods are not ML models as the author's state: "While existing approaches address unlearning for classification and generative models, they overlook an important category of machine learning models: contrastive learning (CL) methods"

**Strengths And Weaknesses:**

Strengths:
1. Well written and visually pleasing paper
2. Paper is likely to be of interest to the unlearning community
3. Good ablation studies to evaluate the proposed method deeply.
4. Contributions beyond simply another unlearning method has broader value to the community. For example I appreciate the formalization of the two-party model (model owner vs data owner).
5. Considerations for both white and black-box evaluations is important and valuable

Weaknesses:
1. I would argue that trying to make the forgetting more visible, via negative-alignment and the AGM, is undesirable. While I appreciate the authors' goal is to provide black-box verification, visible unlearning leaks that the user's information was once in the training data. Sekhari [1] states the goal of unlearning is that with high probability, an observer cannot differentiate between the two cases (i) the model is trained on the set Dt and then a set Du is deleted by the unlearning algorithm and (ii) the model is trained on the set Dr and no points are deleted thereafter by the unlearning algorithm. The proposed negative alignment may violate this, and existing methods that aim to minimise information leakage will likely break the proposed evaluation approach. As a positive, AGM could also be an interesting metric to evaluate this property in the contrastive setting.
2. The proposed method does have a speed trade-off for a relatively marginal performance gain.
3. Missing some related literature on unlearning for contrastive models (e.g. [2]). Further, I think there are other methods that may generalise well to the contrastive setting [3,4,5]. [4] could  use positive pairs as pseudo-labels, and [3,5] are both label-free and would work out of the box.
4. While not a major issue, tuning 3 hyper-parameters is a demanding task.

[1] Sekhari, Ayush, et al. "Remember what you want to forget: Algorithms for machine unlearning." Advances in Neural Information Processing Systems 34 (2021): 18075-18086.

[2]. Wang, Zixin, Bing Mi, and Kongyang Chen. "EncoderMU: Machine Unlearning in Contrastive Learning." International Conference on Security and Privacy in New Computing Environments. Cham: Springer Nature Switzerland, 2023.

[3]. Foster, Jack, Stefan Schoepf, and Alexandra Brintrup. "Loss-free machine unlearning." ICLR Tiny Paper (2024).

[4]. Warnecke, Alexander, et al. "Machine unlearning of features and labels." Network and Distributed System Security Symposium (NDSS) (2023).

[5]. Foster, Jack, et al. An Information Theoretic Approach to Machine Unlearning. Transactions on Machine Learning Research, 2025, https://openreview.net/forum?id=t1utIThKHD.

---

> ### Author Response · Authors · 2025-05-27
>
> We greatly appreciate the reviewer's thoughtful comments and valuable suggestions, which have contributed to strengthening our work. We respond to your feedback below and look forward to any further input:
>
> [**W1: Visible Unlearning and Privacy**] The reviewer raises an important point about potential privacy implications. We acknowledge that from a differential privacy perspective, visible changes from AC could theoretically enable membership inference attacks by observing differences in model behavior compared to retraining, as validated in our AGM analysis (Figure 8, Appendix B.7). However, we argue that such privacy leakage poses minimal practical risk because:
>
> - *Limited data access*: AGM calculations require access to the private unlearn data, which is typically only available to data owners and model developers
> - *No baseline availability*: In practice, the retraining process is not performed, and retrained models are not released, eliminating the comparison baseline needed for inference attacks
> - *Restricted adversary capabilities*: An adversary would need simultaneous access to both the unlearned model and reference data/models, which is unlikely in real deployment scenarios
>
> In summary, while the theoretical privacy concern is valid, the practical barriers to exploitation make such risks negligible in realistic threat models where adversaries lack access to the necessary private information for meaningful membership inference.
>
> We have included the above discussion in Appendix B.8.
>
> [**W2: Speed Trade-off**] We acknowledge the running time trade-off introduced by AC. To further examine the practicality of AC, we have added Table 13 in Appendix B.6, which provides detailed runtime and GPU memory usage across models ranging from ResNet-18 to ResNet-152. Our results demonstrate that AC introduces minimal computational overhead even for the largest models tested.
>
>
> [**W3: Missing references**] We thank the reviewer for the valuable reference suggestions and have incorporated them into our revised manuscript.
> Regarding [2]: While this work addresses a related problem in contrastive unlearning, it differs significantly from our approach in several key aspects. Notably, [2] does not provide a comprehensive evaluation framework or establish strong baseline comparisons. Additionally, their experimental setup uses suboptimal contrastive learning models (achieving only 50-70% test accuracy with SimCLR and MoCo on CIFAR-10), making direct performance comparisons challenging. Our work addresses these limitations by providing both a complete evaluation framework and state-of-the-art baseline performance.
> Regarding [3-5]: We agree that these unlearning algorithms represent important contributions to the field and could potentially be integrated within the MUC framework. We have added a discussion of these methods and identified their integration as promising future work to further expand the scope of our evaluation framework.
>
> [**W4: Hyperparameter Tuning:**] We emphasize that while we introduce three hyperparameters for mathematical generality, only one requires tuning in practice.
>
> Importantly, AC achieves state-of-the-art results with minimal tuning—simply setting $\alpha=\gamma=1$ and using the dataset-determined $\varepsilon$ delivers competitive performance across all our experiments. This demonstrates that AC is not critically dependent on extensive hyperparameter optimization.

---

### Review · Reviewer_PgdP · 2025-05-18

**Summary Of Contributions:**

The paper studies machine *un*learning in the context of contrastive learning models. The goal is to take a machine learning model trained using a contrastive learning algorithm like MoCo, SimCLR, or CLIP, and remove knowledge about a subset of its training data from the model. The exact solution to consider for this task is to re-train the full model on the original dataset with the unlearning samples excluded. The authors study the problem of designing approximate unlearning methods that reach the properties of such an exact model with considerable less compute, and discuss how to evaluate such models.

The central idea is to adopt the contrastive learning objective, which can be split into an alignment and uniformity part (Wang & Isola). On the "retain" portion of the data, the CL objective is minimized as before. Added to this is an adapted CL objective where an additional alignment loss with flipped sign is introduced. A weighted sum of both losses is minimized.

The authors then study the performance of this method against various baselines on CIFAR-10, -100, SVHN for unimodal, and MS-COCO for CLIP.

**Audience:**

Yes

**Broader Impact Concerns:**

The paper presents a method which claims to evaluate the performance of unlearning, which is a legally relevant topic. This makes it interesting and high-impact as a contribution, but also mandates to accurately reflect on the limitations of the algorithms. In particular, I would request the authors to add a section of failure cases of their proposed method, and how to diagnose the respective metrics. I outlined some if these suggestions already above.

**Claims And Evidence:**

No

**Requested Changes:**

I would advise to re-work the results section. Right now, the authors present a large amount of tables and plots with various metrics, without much insightful interpretation in the text. The main story is along the way of beating the state of the art on metrics the authors previously critizied as not meaningful. I see an opportunity here to actually deliver on the motivaton outlined before, e.g., showing that with MUC, it is actually possible for the data owner to meaningfully verify the unlearning procedure. When re-writing, I would keep this in mind.

As an alternative option to some of the points, the authors might opt to write an extensive Limitations section, and re-write some of their earlier motivation.

Please find a list of comments, questions and suggestions below:

1. p.2, "Unfortunately, direct adaptations of existing unlearning appraoches are unsatisfactory [...]": Is this an insight from the literature? If so, which paper(s)? If not, where is this shown in the paper later on?
2. p.3 "Contrastive Learning", "(usually without labels)": I would not enter this debate here; especially the CLIP example is essentially on paired data, I would advise to simply drop the remark, it does not change the story
3. $p^x$ before Eq. 1 is not used in Eq. (1)
4. Eq. (1) omits the fact that many models, e.g. SimCLR, train both an encoder and some kind of projection head on top. This could be minimally noted in the text in case the equation should be kept simple.
5. p.6., **"Exact unlearning on MoCO"**: The paragraph is very anecdotal. Could you transform this into a result/forward reference it? I think it is quite an important statement to make, and I would evaluate this statement properly with a full experiment which shows that FS fails to distinguish the exact unlearned and base models. If this is the case, it would also essentially show that this metric is not useful at all in the further evaluation, but e.g. Table 7 still shows an evaluation (incl. bolded numbers) on it.
    - As an actionable proposal, a convincing argument would be to consider the RS metric between the baseline and re-trained model, compute a distribution, and test the null hypothesis: mean(RS)=0. This could be a very convincing part of the motivation. I would encourage the authors to look into this and make the statement statistically more precise.
6. Eq. 3 and 4, is it correct that the negatives are sampled across the whole distribution $p_d$ (vs. $p_r$)?
7. Eq. 4 the terms of "positive alignment" and "negative alignment" are a bit confusing. With this naming, the "positive alignment" is actually *not* the term that aligns the positive samples (in the sense of Wang & Isola), while the "negative alignment" is. I would re-consider the naming here.
8. The Eq. in "Taking evaluation into account" is not labeled; it is also every simple to confuse $p_u^x$ and $p_u^+$ when reading the equation. Could you also clarify what the difference between $p_u^x(x,y)$ and $p_u(x)p_d(y)$ is?
9. p.7. "randomly forget": Did you study what happens when the forgetting request is more targeted, e.g. dropping out a part of a whole class, searching for a particular keyword, removing images associated to a particular photographer?
10. p.8, "we tune $\beta$ for the best performances" -> how is this tuning done in practice, what is the metric?
11. Table 1: Where is "Alignment Calibration (AC)"? In the text it is mentioned, in the table not. Is that "Ours"?
12. The abstract claims, "Alignment calibration approximates exact unlearning". While Table 1 and Table 4 show improved performance, it is unclear why AC approximates this "more" than the baselines. Are any claims possible in which fraction of cases (%) you are able to correctly certify that unlearning was successful (= the model becomes indistinguishable from the retrain model)?
13. What is the standard deviation of the results in the table? Especially with the point raised in Q6, it seems important to report the confidence intervals/standard errors in the table. I would consider re-layouting, and removing the (...) values in favor of standard deviation. A statistical test would be a plus.
14. FS and downstream metrics are (I think) never compared directly. The abstract claims "Enables visualization of unlearning effects through black-box evaluation" --- where is this done/discussed?
15. Figure 2, Fine-tune, NegGrad, l1, AC essentially looks like the same distribution. Still the text claims, "exhibits a more signficant unlearning effect". Can you either run a statistical test showing this, or drop the claim? I think the figure essentially shows what you alluded to in the text: Standard deviation of this statistic is so large that it is essentially impossible to get a meaningful message out of it with this evaluation protocol.
16. Figure 4 should be converted to a different plot type, or the y axis should start at 0. I would recommend a boxplot w/ error bars or stripplot, or use a table.
17. Table 8 could be made clearer, there is enough space to put labels above the "w/ + w/" column for either readability

**Strengths And Weaknesses:**

The paper (up to the results section) is overall well written, and the motiviation and approach is clearly outlined. The proposed framework is targeted at contrastive learning in general and in that sense could be quite impactful in the real-world. At the same time, it is intuitive and easy to apply (assuming access to the training dataset).

The biggest weakness of this paper is the depth of the empirical analysis, as well as the mismatch between claims about evaluation and what the authors actually did. Broadly speaking, in Section 3.1 the setup is nicely introduced as a process between model owner and data owner, metrics are introduced (and criticised) but then in the results, the authors do not deliver much alongside this story.

I believe this is mostly an issue with writing and result presentation, and less about running fundamentally new experiments --- I think this part is done well.

In general, the most important result I anticipated while reading everything up to the results is that the data owner would be able to get a model unlearned with baselines and the new AC method, and could then evaluate if the model successfully unlearned the task. A setup for this could be some kind of hypothesis test where the null hypothesis is that the model behaves like the original model, and H0 needs to be rejected with at some p-value to denote that the given model indeed unlearned the task.

The way the existing metrics are critized ("Exact unlearning on MoCo") alludes to this.

Instead, the authors evaluate their new model on a set of metrics and show "continuous" improvements over previous methods. From the data owner perspective, "[the data owner] should find explicit evidence that unlearning has indeed been performed and the output is as desired"; from the metrics presented, it is unclear to me what the contribution towards this goal is. Does one of the "new" metrics perform better than the "old" (e.g. RS metric) towards this goal (i.e., is the ranking of algorithms different)? How would the data owner verify that the unlearning was indeed successful?

### Claims & Evidence

Looking at the claims in the abstract,

1. Alignment calibration achieves SOTA
2. Alignment calibration approximates exact unlearning
3. Enables visualization of unlearning effects through black-box evaluation

I think that (1) is sufficiently addressed. The authors show various metrics and achieve good results on these metrics. On the flip-side, I do not see the claim for (2) and (3), but might miss something. Could you concisely outline which experimental results support (2) and (3)? Please see the requested changes for additional comments on this (from Q12 onwards).

---

> ### Author Response · Authors · 2025-05-27
>
> We sincerely thank the reviewer for the detailed feedback and constructive suggestions that have helped us significantly improve our paper. We address your comments below.
>
> [**Weaknesses: Claims & Evidence**]
> We would like to first address the reviewer's concern regarding the perceived mismatch between our claims and empirical results. We believe this stems from a misunderstanding of how our experimental design directly implements and validates the MUC framework.
>
> Our experimental evaluation comprises two complementary components that directly correspond to our framework and claims:
>
> (1) **White-box evaluation** (Section 5.2, reflecting Section 3.3): This component evaluates and compares approximate unlearning algorithms using metrics accessible to model owners, directly supporting our claim that "Alignment calibration approximates exact unlearning" under this evaluation. The white-box results demonstrate AC's effectiveness relative to other methods and retraining baselines.
>
> (2) **Black-box evaluation** (Section 5.3, reflecting Section 3.4): This component provides evaluation tools for data owners who lack internal model access, directly supporting our claim that we "Enable visualization of unlearning effects through black-box evaluation." These results show how individuals can assess unlearning success without white-box access.
>
>
> With this general setup, we believe many of the reviewer's concerns can be mitigated, including:
>
> > The main story is along the way of beating the state of the art on metrics that the authors previously criticized as not meaningful.
>
> This characterization misrepresents our position and contributions. Our white-box evaluation results are not our "main story" but rather one component of our comprehensive framework. Importantly, we never claim white-box evaluation metrics are "not meaningful." Instead, as noted in Section 3.4, we point out that data owners cannot easily perform such evaluations due to limited model access, and that relying on FS alone can be misleading without additional context
>
> > From the data owner's perspective, "[the data owner] should find explicit evidence that unlearning has indeed been performed and the output is as desired"; from the metrics presented, it is unclear to me what the contribution towards this goal is.
>
> This concern is precisely why we propose AM and AGM metrics in Section 4 and provide their visualization in Section 5.3. These black-box metrics enable data owners to assess unlearning effectiveness without requiring internal model access, directly addressing the identified need for accessible evaluation tools.
>
> **In summary**, our empirical evaluation directly aligns with and validates the framework presented in Sections 3 and 4, rather than deviating from our main contributions. We hope the above clarification addresses the reviewer's concerns about the perceived mismatch between our claims and results. If any concerns remain, please let us know and we would be happy to make further adjustments to improve the paper's clarity.
>
> Next we address your detailed comments in the "requested changes" section:
>
> [**Requested changes**]:
> 1. *Direct adaptations of existing unlearning approaches are unsatisfactory.*:
>
> This insight is not derived from existing literature, as we are the first to systematically adapt general unlearning algorithms to the contrastive learning setting. Instead, as noted on Page 5 (paragraph 1), this key observation stems from our empirical findings—specifically, the baseline method performance reported in Section 5 for both black-box and white-box evaluations. These empirical results revealed the limitations of direct adaptations and served as the primary motivation for designing our novel AC method.
>
> 2. *Contrastive Learning (usually without labels)*:
>
> We agree that "label" could be misleading here, and we have removed the note.
>
> 3. $p^\times$: Thank you for the note, we have deleted this notation.
>
> 4. *$g$ may cover a projection head*. Thank you for the suggestion. We have added a footnote on Page 3 to cover such cases.

---

> > ### Author Response · Authors · 2025-05-27
> > **Rebuttal (Continued)**
> >
> > 5. *Exact unlearning on MoCo*
> > - [FS is not useful at all]: We emphasize that the FS score is unreliable *only under limited access conditions* (black-box scenarios) and remains an important evaluation metric for model owners conducting white-box evaluation and algorithm selection.
> > - [Hypothesis test]: Using a fixed random seed of 10, we pretrain a MoCo encoder $g_1$ on the entire CIFAR-10 training set. For exact unlearning, we retrain an encoder $g_2$ after removing 10\% of the data, with the same random seed. To simulate a scenario where the model owner attempts to deceive the unlearning process, we assume that instead of retraining, the original encoder $g_1$ is replaced with another encoder $g_3$ that is pre-trained on the entire training data with a different random seed of 11. In this setting, the null hypothesis $H_0$ (no unlearning) corresponds to the deceptive case in which $g_1$ is replaced by $g_3$. The alternative hypothesis $H_1$ represents the exact unlearning case where $g_1$ is replaced by $g_2$. For both hypotheses, we calculate the FS for every unlearn sample and get the mean $\mu$ and the standard deviation  $\sigma$ across the 4500 unlearning images. The resulting statistics are: $H_0: (\mu_0=0.0353, \sigma_0=0.0575)$;  $H_1: (\mu_1=-0.0026, \sigma_1=0.0587)$. Assuming Gaussian distributions and a data owner possesses $N$ images, we perform a t-test to evaluate the $p$-value for distinguishing between $H_0$ and $H_1$:
> >
> >     |     $N$    |     5    |     10    |     15    |     20    |
> >     |---|---|---|---|---|
> >     |     $p$-value    |     0.3322    |     0.1617    |     0.0847    |     0.0459    |
> >
> >     From this table, we observe that when the data owner holds fewer than 15 images, it becomes difficult to statistically detect that the model owner has cheated the unlearning process.
> >
> > 6. *Equation 3 and 4:* Yes, the equations are correct. We sample negatives from the entire distribution because this allows for efficient batch computation.
> >
> > 7. *Positive alignment and negative alignment*: Both terms are consistent with the alignment definition from Wang & Isola, where we specify alignment for positive pairs and negative pairs within the unlearn set, respectively.
> >
> > 8. *Taking evaluation into account*: $p_u^\times (x,y)$ represents the negative pairs from the unlearn samples, while $p_u(x) p_d(y)$ specifies that $x$ is an unlearn sample and $y$ is sampled from the entire distribution.
> >
> >
> > 9. *Targeted forget*: While the reviewer raises a very interesting setting, our work focuses on random forgetting as a first step. We believe targeted forgetting—particularly understanding the difficulty of removing different samples in the contrastive learning setting—is very important. We plan to extend existing studies (e.g., [1]) to contrastive learning in future work.
> >
> > [1] Zhao et al., What makes unlearning hard and what to do about it, NeurIPS 2024
> >
> >
> >
> > 10. $\beta$: We set $\alpha=\gamma=1, \varepsilon= |D_{unlearn}|/|D_{retain}|$ and only tune $\beta$ to achieve the lowest "Average Gap" as reported in the tables.
> >
> > 11. Yes, "ours" represents "AC" in all tables. We have updated this notation consistently throughout the paper for clarity.
> >
> > 12. *Approximate exact unlearning*: Regarding the claim that "Alignment calibration approximates exact unlearning," AC demonstrates closer approximation to exact unlearning (retraining) than baselines through several key metrics:
> > - Tables 1 and 4 show that AC consistently achieves smaller gaps compared to retraining across multiple metrics (FS, MIA, etc.), indicating better approximation of the exact unlearning objective. The "Average Gap" metric specifically quantifies how closely each method approximates retraining performance.
> > - AC's design specifically accounts for contrastive learning properties—unlike generic unlearning methods adapted to this setting, AC explicitly addresses both positive and negative alignment within the contrastive framework, leading to behavior that more closely matches retraining.
> >
> > Regarding theoretical guarantees, we focus on approximate unlearning following established literature, which evaluates methods using empirical metrics rather than formal guarantees.
> >
> > 13. *Standard deviation*: We have indeed included these statistics in Table 11 and Table 12.
> >
> > 14. (1) FS and downstream metrics represent different evaluation dimensions and are presented in separate tables for clarity, as they operate on quite different scales; (2) We visualize the unlearning effects in Figures 3, 6, 7.

---

> > > ### Author Response · Authors · 2025-05-27
> > > **Rebuttal (Continued)**
> > >
> > > 15. We propose that data owners can calculate the AGM and examine the negative alignment from off-diagonal entries, where lower negative alignment indicates more significant unlearning effects.
> > >
> > > Figure 2 demonstrates that our AC method achieves lower average negative alignment with similar variance compared to baselines. For example, AC's mean and standard deviation are (-0.053, 0.03), while NegGrad's are (-0.037, 0.029), indicating that AC provides a more reliable evaluation tool.
> > >
> > > Assuming Gaussian distributions and considering a toy anomaly distribution of (0, 0.01), AC achieves an AUC of 0.964 compared to NegGrad's 0.886. This demonstrates AC's superior discriminative capability for detecting unlearning effects.
> > >
> > > We acknowledge that we cannot access a real anomaly distribution since the true nature of how model owners might circumvent the unlearning process remains unknown in practice.
> > >
> > > 16. Figure 4 reports the ratio FS(RT):FS(AC), which remains near the baseline value of 1.0. Therefore, we use a y-axis range from 0.9 to 1.1 to clearly visualize the variations around this baseline. Since FS(RT) and FS(AC) represent already-averaged values, error bars cannot be meaningfully reported for their ratio.
> > >
> > > 17. We have made updates to clarify the notations in the table.

---

> > > > ### Comment · Reviewer_PgdP · 2025-05-28
> > > > **Re: (2) Enable visualization of unlearning effects through black-box evaluation.**
> > > >
> > > > Thank you very much for the clarifications. Before replying in full, I would like to address a potential remaining misunderstanding head-on. My key confusion is around the white-box vs. black-box evaluation terminology and proposed metrics:
> > > >
> > > > - According to the definition in the paper, FS is a black-box score, is that correct? Specifically, the data owner could evaluate this metric, but now you show that this requires at least 20 samples to reach a 5% signficance level.
> > > > - AM and AGM are then extensions of FS, specifically the main diagonal of AGM is related to FS.
> > > >
> > > > If this is the case, what I am missing right now is a comparison that AM and AGM are more effective tools for the model owner to "rank" models compared to FS. Specifically, FS is a scalar test-statistic. How do you perform the same test with AM or AGM beyond a qualitative comparison and visualization? Does the metric in Figure 2 serve this purpose? If so, how does this metric compare to FS at the same budget of available images (4,500)?
> > > >
> > > > What is the performance of AM and AGM if you repeated the test from the section "Exact unlearning on MoCo" on these metrics?
> > > >
> > > > In Figure 3, how do these matrices look for other models? Would you obtain a different ranking of the models by AM or AGM?

---

> > > > > ### Author Response · Authors · 2025-05-30
> > > > >
> > > > > Thank you for your further comments! Here we address your remaining concerns below:
> > > > >
> > > > > > According to the definition in the paper, FS is a black-box score, is that correct?
> > > > >
> > > > > FS can be used for both white-box and black-box evaluation according to its definition, but we demonstrate that it is unreliable for the latter when the sample size is small.
> > > > >
> > > > > > AM and AGM are then extensions of FS, specifically the main diagonal of AGM is related to FS.
> > > > >
> > > > > This is correct. We leverage negative pairs in addition to positive pairs to enable more accessible evaluation for data owners.
> > > > >
> > > > > > a comparison that AM and AGM are more effective tools for the model owner to "rank" models compared to FS.
> > > > >
> > > > > The reviewer raises an important aspect that our paper previously lacked. To address this, we conducted the same hypothesis test with an identical budget (4500 samples) using negative alignment and compared it with FS under AC unlearning. We invite you to review our updated Section 5.3 and Table 9 (marked in blue), where negative alignment exhibits superior performance compared to the FS score under AC, highlighting the benefits of our Section 4 design.
> > > > >
> > > > > Additionally, we acknowledge that our current analysis considers one typical deceptive case for the null hypothesis, where the model owner cheats by replacing the encoder without performing unlearning. In future work, we will investigate additional failure cases where malicious model owners deliberately design cheating methods to deceive the proposed metrics and develop corresponding mitigation strategies (relevant discussion added in Section 6).
> > > > >
> > > > >
> > > > > > In Figure 3, how do these matrices look for other models? Would you obtain a different ranking of the models by AM or AGM?
> > > > >
> > > > > We investigate different unlearning methods (MoCo & CLIP) and models (ResNet-50) in Figures 6 and 7 in Appendix B.2, where our observations regarding AC's superiority remain consistent.

---

> > > > > > ### Comment · Reviewer_PgdP · 2025-06-14
> > > > > >
> > > > > > Dear authors, thanks for adding this. What I am still missing in Table 1 and Table 9 is the control. You have three models now,
> > > > > >
> > > > > > (1) The original model
> > > > > > (2) The original model trained with a different seed
> > > > > > (3) A model performing exact unlearning
> > > > > >
> > > > > > Here is the question Table 1 and 9 still not address, unless I am missing something: How well can a data owner distinguish whether a candidate model (*) is (2) or (3) from a limited set of images N, for FS vs. the negative alignment gap? Are the p-values in Table 9 computed based on all three models (1)-(3) or are these only for comparing (1) vs. (3)?
> > > > > >
> > > > > > A way to re-phrase my question: Assume the model owner provides a model, and the data owner makes a decision to acknowledge the successful unlearning. Is the metric in Table 9 then the actual probability that the model owner choses to accept the unlearning, for the case where the provided model is actually just re-trained with a different seed?
> > > > > >
> > > > > > The way how I read your methods right now is that Table 9 reports the p-value computed by running a t-test on a set of images, and you report this p-value for comparing the mean/standard deviation of (1) to (3) and report this. But did you verify that this corresponds to the actual error rate of the model by comparing (1) and (2) vs. (1) and (3)?

---

> > > > > > > ### Author Response · Authors · 2025-06-14
> > > > > > >
> > > > > > > Thank you for the further comments!
> > > > > > >
> > > > > > > We must clarify that the unlearned model reported in Table 9 is based on our proposed Alignment Calibration (AC) method rather than exact unlearning. Thus, model (3) represents an AC-unlearned model.
> > > > > > >
> > > > > > > Consider a data owner with $N$ images who has access to the original model (1) and a current model (\*) that could be either model (2) or (3). Our goal is to demonstrate the superiority of the proposed negative alignment metric in distinguishing between the AC-unlearned model and the replacement pretrained model. Given $N$ images and the two models (1) and (\*), the data owner can compute $N(N−1)/2$ negative alignment values to approximate the distribution using the mean and standard deviation. We report the p-value of a t-test for distinguishing between these two distributions.
> > > > > > >
> > > > > > > We believe the "actual error rate" mentioned by the reviewer refers to an ideal scenario where the two distributions need not be approximated independently by the data owner, but are instead provided as known priors, allowing verification using the $N$ samples. However, such an ideal scenario is impractical because the potentially malicious model provider—who performs unlearning and is assumed capable of cheating—has no incentive to supply accurate prior distributions. This challenge of black-box evaluation for individual data owners in the presence of potentially untrustworthy model providers precisely motivates our work.

---

### Author Response · Authors · 2025-05-27
**Summary of Changes**

We thank all reviewers for their dedication to reviewing our submission. We have addressed your concerns below and modified the paper accordingly. All modifications are marked in teal, and we provide a summary of changes below:
- We added a discussion of additional references in a footnote on Page 2;
- We removed the definition of $p^{\times}$ on Page 3, which was not used in the remaining paper;
- We added a footnote on $g$ to include the architectural design of projection heads on Page 3;
- We improved the paragraph "Exact unlearning on MoCo" with a hypothesis test on Page 6;
- We changed "Ours" to "AC (Ours)" in all tables;
- We improved the presentation of Table 9;
- We added a "Limitations and Future Work" paragraph in Section 6, specifying future directions on (1) other SSL methods; (2) including other label-free unlearning algorithms in MUC; (3) MUC with theoretical guarantees; (4) unlearning difficulty in contrastive learning;
- We added Appendix B.6 to discuss time and memory consumption when scaling to larger models;
- We discussed privacy concerns in Appendix B.7;
- We extended ablation studies on hyperparameter tuning in Appendix B.8.

---

### Decision · Action_Editor_Vraj · 2025-07-16

**Recommendation:** Accept as is

**Additional Comments:**

This paper is on the topic of unlearning for contrastive learning methods. They propose a novel method, called Alignment Calibration, that is based on the InfoNCE objective, and evaluation metrics based the divergence of alignment of augmented outputs before and after unlearning. They also present a conceptual framework that captures the interactions between a model owner and data owner, where the model owner can perform white box evaluation of unlearning (given access to the model weights) but the data owner may be limited to black box access. They demonstrate that, on various evaluation metrics, their method outperforms the baselines.

The reviewers found the paper "well written" (Reviewer PgdP, Reviewer ownh), has a "clear problem definition" (Reviewer HyuW), "the motiviation and approach is clearly outlined" (Reviewer PgdP), "could be quite impactful" (Reviewer PgdP). The proposed algorithm is "Novel and Well-Motivated" (Reviewer HyuW).
The reviewers also found that the authors conducted "good ablation studies to evaluate the proposed method deeply" (Reviewer ownh), their experiments consider both white and black-box metrics which is "important and valiable" (Reviewer ownh). The authors also worked very hard during the rebuttal to add more experiments, statistical tests, make various clarifications and discuss limitations (see above section for more details).

**Audience:**

Yes

**Audience Explanation:**

Unlearning is an important area receiving increasing attention from the community recently. This paper represents the first to the best of my knowledge study of unlearning in the context of contrastive learning algorithms, including appropriate baselines, presenting the novel algorithm and discussing evaluation metrics for this problem.

**Claims And Evidence:**

Yes

**Claims Explanation:**

This paper is on the topic of unlearning for contrastive learning methods. They propose a novel method, called Alignment Calibration, that is based on the InfoNCE objective, and evaluation metrics based the divergence of alignment of augmented outputs before and after unlearning. They also present a conceptual framework that captures the interactions between a model owner and data owner, where the model owner can perform white box evaluation of unlearning (given access to the model weights) but the data owner may be limited to black box access. They demonstrate that, on various evaluation metrics, their method outperforms the baselines.

The main claims of the paper are that Alignment Calibration achieves SOTA compared to the baselines for unlearning in contrastive learning, and according to the (comprehensive) metrics considered. These claims are sufficiently substantiated through extensive experiments and ablation studies, as recognized by the reviewers. Reviewer PgdP had substantial productive discussions with the authors about the presentation of the results in order to best demonstrate that the claims are satisfied. Since then, the authors have added additional experiments and hypothesis testing to strengthen the experimental evaluation. Generally, evaluation of unlearning is a very challenging area (and an open problem in and of itself) but, after discussions with reviewers PgdP and ownh, the authors have updated the paper to acknowledge and discuss limitations. For example, Reviewer ownh brought up that, while enabling the data owner to "visualize" the effect of unlearning on their data can be desirable for the data owner to confirm that unlearning has succeeded, it can be problematic if it allows (some types of) attackers to also infer which data has been unlearned, causing a privacy breach. The authors have discussed that possibility in their updated manuscript and whether / when it is realistic.